# Amyloid-β fibrils accumulated in preeclamptic placentas suppress cytotrophoblast syncytialization

Kaho Nishioka[1], Midori Ikezaki[2] , Naoyuki Iwahashi[1] , Miyu Arakawa[2], Momo Fukushima[2], Noa Mori[2], Mika Mizoguchi[1], Yuko Horiuchi-Tanizaki[1], Megumi Fujino[1], Takami Tomiyama[3], Yoshito Ihara[2] , Kenji Uchimura[4] , Kazuhiko Ino[1], Kazuchika Nishitsuji[2,4] 

**Cerebral deposition of fibrillar amyloid-β (Aβ) is a pathological hallmark of Alzheimer's disease. Although Aβ is present in human placentas and accumulates in preeclamptic placentas characterized by poor placentation, the production and role of Aβ in the human placenta remain unclear. Because hypoxia in mid-to-late pregnancy is a risk for preeclampsia, we found that levels of hypoxia-inducible factor 1-α and β-secretase (BACE-1) increased concurrently with placental Aβ deposition in late-stage preeclamptic placentas. We also found that a human cytotrophoblast (CTB) model, BeWo cells, actually produced Aβ species and that hypoxia increased Aβ production and BACE-1 protein levels. Aβ42 fibrils inhibited CTB syncytialization, a critical step in maintaining pregnancy, by inducing loss of membrane localization of cell–cell adhesion molecules. Primary human CTBs confirmed these observations. Taken together, our results suggest that increased Aβ production in CTBs by hypoxia may lead to the formation of Aβ fibrils, which inhibit syncytiotrophoblast formation and are detrimental to pregnancy. Thus, our results reveal the novel role of Aβ fibrils in the pathogenesis of preeclampsia.**

## Introduction

Preeclampsia (PE) is one of the most severe pregnancy-specific disorders associated with hypertension and proteinuria. Approximately 7–10% of pregnancies manifest PE, which may result in high maternal and fetal morbidity and mortality (Khan et al, 2006; Wallis et al, 2008; Duley, 2009; Abalos et al, 2013). At this time, no cure other than delivery of the fetus exists for PE, which makes PE the leading cause of iatrogenic preterm birth (Phipps et al, 2019). Although the etiology of PE is not yet fully understood, poor placentation has been implicated in PE pathophysiology (Aplin et al, 2020; Chappell et al, 2021). The abnormal deposition of misfolded proteins, such as transthyretin and Thr231-phosphorylated *cis*-P-tau, has been implicated in the etiology and pathology of PE (Cheng et al, 2021, 2022; Jash et al, 2023; Kalkunte et al, 2013). In addition, several studies reported deposition of aggregated amyloid β (Aβ) peptides in PE placentas (Buhimschi et al, 2014; Cater et al, 2019; Cheng et al, 2021). Aβ peptides, which deposit in the brains of patients with Alzheimer's disease (AD), are produced by the sequential cleavage of amyloid precursor protein (APP) by β-secretase 1 (BACE1) and γ-secretase. Aβ levels in the brain are determined by the balance between Aβ production and clearance (Selkoe & Hardy, 2016). Thus, an imbalance between Aβ production and clearance leads to increased Aβ levels and Aβ aggregation to form toxic Aβ aggregates. APP is widely expressed throughout the body, including the placenta, and can be processed by BACE1 and γ-secretase to produce Aβ in the placenta (Mattson, 2004; Buhimschi et al, 2014). Association between hypertensive disorders in pregnancy including PE and dementia and the involvement of aggregation of proteins in PE have been documented (Basit et al, 2018; Cheng et al, 2016; Olie et al, 2024; Schliep et al, 2023), which supports that PE is a placental proteinopathy. Aβ aggregates have been established as toxic to neurons (Selkoe & Hardy, 2016); however, exactly how Aβ and Aβ aggregates affect placental cell functions is unknown.

Cytotrophoblasts (CTBs) are epithelial stem cells in the human placenta that differentiate into two major placental cell types: extravillous trophoblasts (EVTs) and syncytiotrophoblasts (STBs) (Bischof & Irminger-Finger, 2005). CTBs undergo continuous syncytialization to form STBs in the outer layers of the floating chorionic villi, which are critical for key placental functions such as fetal nutrition, gas exchange, and protection and placental hormone production (Kliman et al, 1986; Eaton & Contractor, 1993; Ogren & Talamentes, 1994). Hypoxia reportedly inhibited CTB syncytialization via a transcription factor complex known as hypoxia-inducible factor (HIF) (Jaremek et al, 2023), and CTB

[1]Department of Obstetrics and Gynecology, School of Medicine, Wakayama Medical University, Wakayama, Japan    [2]Department of Biochemistry, School of Medicine, Wakayama Medical University, Wakayama, Japan    [3]Department of Translational Neuroscience, Osaka Metropolitan University Graduate School of Medicine, Osaka, Japan    [4]Unité de Glycobiologie Structurale et Fonctionnelle, UMR 8576 CNRS, Université de Lille, Villeneuve d'Ascq, France

Correspondence: nishit@wakayama-med.ac.jp

**Table 1. Clinical information of the study population.**

| Case no. | Maternal age (year) | Maternal BMI | Gestational age (week) | Neonatal weight (g) | Placental weight (g) | Early or late onset | IUGR[a] |
|---|---|---|---|---|---|---|---|
| Normal_#1 | 34 | 18.7 | 37 | 2,759 | 530 | | – |
| Normal_#2 | 24 | 25 | 37 | 2,577 | 472 | | – |
| Normal_#3 | 33 | 23.6 | 35 | 2,739 | 614 | | – |
| Normal_#4 | 26 | 22.3 | 37 | 2,549 | 394 | | – |
| Normal_#5 | 38 | 20.2 | 33 | 2,154 | 784 | | – |
| PE_#1 | 36 | 22.3 | 33 | 1,605 | 376 | Early onset | – |
| PE_#2 | 25 | 24 | 32 | 1,849 | 364 | Early onset | – |
| PE_#3 | 23 | 23.9 | 35 | 2,446 | 608 | Late onset | – |
| PE_#4 | 38 | 25.1 | 33 | 2054 | 458 | Early onset | – |
| PE_#5 | 35 | 26.6 | 31 | 1,015 | 247 | Early onset | + |

[a]IUGR, intrauterine growth restriction.

syncytialization was reduced in PE placentas (Costa, 2016), which indicated a role of hypoxia in poor placentation and PE pathology.

Hypoxia associated with cerebral ischemia and stroke is a strong risk for the development of late-onset AD (Kokmen et al, 1996; Tao et al, 2024; Tatemichi et al, 1994). Hypoxia associated with cerebral ischemia and stroke is thought to increase Aβ production by inducing the expression of BACE1 (Sun et al, 2006; Zhang et al, 2007). Because PE and AD share hypoxia as a common factor that is implicated in the pathogenesis, we hypothesized that hypoxia may increase Aβ production in preeclamptic placentas, leading to the formation of toxic fibrillar Aβ and inhibiting STB formation. The main Aβ peptide species are Aβ40 and Aβ42, and Aβ42 is more predisposed to aggregation than is Aβ40 (Tanzi & Bertram, 2005). Indeed, genetically modified mice that generate Aβ42, but not Aβ40 alone, developed amyloid plaques (McGowan et al, 2005). Therefore, chronic hypoxia in the dysplastic placenta may increase Aβ production and subsequent formation of toxic Aβ aggregates. Here, we report Aβ deposition in preeclamptic placentas and that hypoxia enhanced BACE1 expression and Aβ production in CTB model BeWo cells. We also show that Aβ fibrils suppressed differentiation of CTB model BeWo cells and primary human CTBs. Our results support the pathological role of Aβ fibrils in poor placentation by interfering with STB formation.

## Results

### Aβ accumulated and hypoxia was enhanced in PE placentas

We performed immunohistochemical analyses of normal and PE placentas in late pregnancy with the β001 anti-amyloid β antibody. The information for each of the cases is shown in Table 1. Of the 5 PE cases, one was a late-onset case and one was an early-onset case diagnosed with intrauterine growth restriction. We used the ProteoStat Protein Aggregation Assay kit for analysis of aggregated Aβ. Aβ peptides are small peptides consisting of 40 to 42 amino acid residues that are released extracellularly after production.

Nondeposited Aβ monomers are not detected by our immuno-histochemical analysis, because these soluble Aβ peptides are spread out in the tissue fluid. Thus, we calculated only merged signals of Aβ and the ProteoStat dye to show aggregated and deposited Aβ peptides in the placenta. Here, we found that there is significant deposition of aggregated Aβ in the villi of five PE cases but not in normal placentas (Fig 1). We also used the RB4CD12 anti-heparan sulfate S-domain antibody as an amyloid/protein aggregate marker, because heparan sulfate S-domains have been shown to co-deposit with amyloid in vivo (Bruinsma et al, 2010; Hosono-Fukao et al, 2012; Iwahashi et al, 2020; Kameyama et al, 2019; Nishitsuji, 2018; Nishitsuji & Uchimura, 2017). Again, we found co-deposition of Aβ with the RB4CD12 epitope in PE placentas (Fig S1).

The transcription factor HIF-1α is activated in response to hypoxia and plays an important role in cell responses and adaptation to hypoxia (Weidemann & Johnson, 2008). Hypoxia and HIF1-α reportedly enhanced BACE1 expression and Aβ production (Sun et al, 2006; Zhang et al, 2007). Because PE placentas exist under hypoxic conditions (Tong et al, 2022), we analyzed the expression of HIF1-α in human normal and PE placentas. In a normoxic condition, HIF1-α is constitutively expressed but degraded via the proline hydroxylation and the subsequent ubiquitination and degradation in the proteasome. Because the proline hydroxylation is oxygen-dependent, hypoxia induces HIF1-α accumulation (Masson & Ratcliffe, 2003). Immunohistochemistry revealed the induction of HIF-1α in PE placentas, suggesting a hypoxic environment consistent with a previous study (Caniggia & Winter, 2002). We also found that BACE1 expression was significantly increased in PE placentas (Fig 2). HIF-1α was mainly induced in STBs and CTBs, whereas BACE1 expression was observed in CTBs and STBs, as well as in stromal tissues in PE placental villi. The current study includes four early-onset and one late-onset PE cases (Table 1). The placentas of patients with early- and late-onset PE are under hypoxic and hypo-perfusion conditions (Soto et al, 2012; Baylis et al, 2024). Our immunohistochemical analysis revealed that early- or late-onset PE placentas were in a hypoxic condition, which may lead to a sustained increase in the

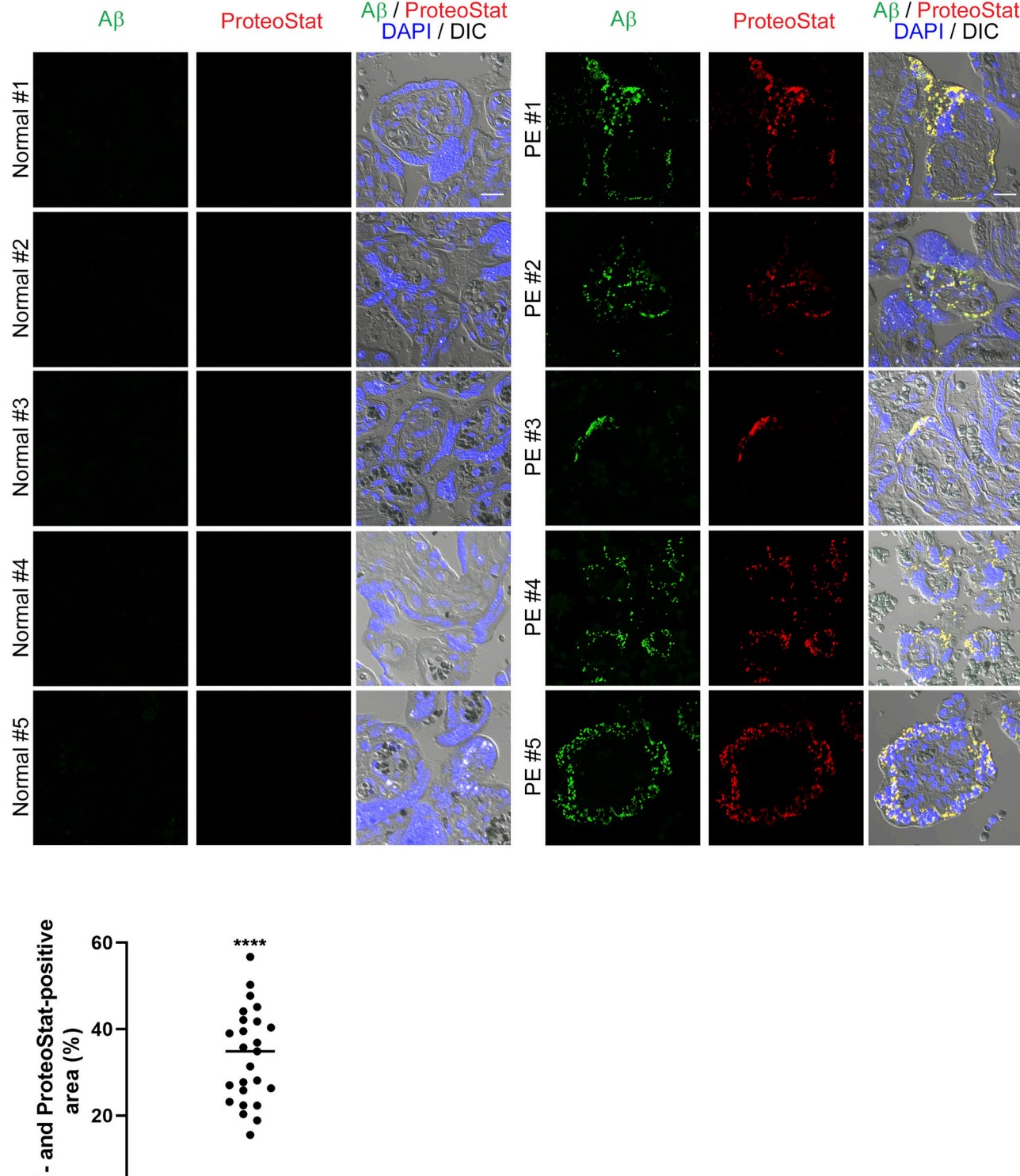

**Figure 1. Aβ deposition in PE placentas.**

Immunohistochemical analysis of normal and PE human placentas from the third trimester. Sections were stained with the β001 rabbit polyclonal anti-amyloid β antibody and the ProteoStat protein aggregation detecting dye. The graph shows the quantification of β001- and ProteoStat-positive area in ROIs that represents aggregated and deposited Aβ. ****$P < 0.0001$. Scale bar, 20 μm.

Source data are available for this figure.

production and local concentration of Aβ peptides and fibrilli-zation of Aβ. Thus, these results support that Aβ deposition may be involved in the both early- and late-onset PE pathologies. Although we detected BACE1 in normal placentas, Aβ did not deposit in these placentas. As mentioned above, nondeposited soluble Aβ peptides in normal placentas are thought to spread out in the tissue fluid and were not detected by our immunohistochemical analysis.

### Hypoxia increased BACE1 levels and Aβ production in CTB model cells

In our study, we used BeWo cells, which are widely used as a model for trophoblast syncytialization (Drewlo et al, 2008; Gauster et al, 2009). Hypoxia treatment increased HIF1-α and BACE1 protein levels (3.8-fold for HIF1-α and 1.5-fold for BACE1, Fig 3A). Prolyl hydroxylation of HIF1-α acts as a signal for the ubiquitin/proteasome-mediated degradation of HIF1 (Schofield & Ratcliffe, 2004). To study the effect of HIF-1α on BACE1 expression, we used the HIF1-α stabilizer roxadustat (Su et al, 2020), which inhibits the prolyl hydroxylation and subsequent degradation of HIF1-α (Hsieh et al, 2007). Roxadustat (0, 5, and 10 μM) significantly increased HIF-1α expression (3.1-fold for 5 versus 0 μM, and 5.5-fold and 1.5-fold for 10 versus 0 and 5 μM, respectively) and BACE1 (1.4-fold for 5 μM and 1.6-fold for 10 versus 0 μM) in BeWo cells (Fig S2). Thus, hypoxia-induced up-regulation of HIF-1α enhanced BACE1 expression in BeWo cells, which may eventually increase Aβ production.

According to our ELISA, Aβ40 production increased 126% under hypoxic conditions compared with Aβ40 production under normoxic conditions, and it decreased, 44% and 34%, after use of the LY2886721 BACE1 inhibitor under both normoxic and hypoxic conditions, respectively (Fig S3). We then analyzed Aβ production in these cells by means of Western blotting with the β001 anti-Aβ N terminus (Nt) antibody (Lippa et al, 1999). Aβ40/42 production in BeWo cells increased 150% under hypoxic conditions compared with that under normoxic conditions (Fig 3B). We also found that the BACE1 inhibitor reduced Aβ40/42 production in BeWo cells under normoxic and hypoxic conditions (40% and 33%, respectively; Fig 3B). We did not observe Aβ oligomers in CM obtained from BeWo cells (Fig S4). Although our ELISA failed to detect Aβ40 in the conditioned medium (CM) of the EVT model HTR8/SVneo cells (Graham et al, 1993), we found that HTR8/SVneo cells produced Aβ peptides and that the Aβ production was enhanced by hypoxia in the immunoblot with the β001 antibody (Fig S5). These results suggested that HTR8/SVneo cells produced much less amount of Aβ than BeWo cells.

### Aβ42 fibrils inhibited syncytialization of CTB model BeWo cells

Hypoxia has been implicated in placentation dysfunction after the formation of spiral arteries. We therefore hypothesized that increased production of Aβ by CTBs and subsequent aggregation of Aβ may have detrimental effects on CTB functions and thereby contribute to defects in placentation and development. Although Aβ40 is generally the predominant species (Burdick et al, 1992), Aβ42 is more prone to aggregation than Aβ40 (Suh & Checler, 2002). Because syncytialization of CTBs is a critical event for placentation,

is maintained until the end of pregnancy, and is reportedly impaired in PE placentas (Costa, 2016), we investigated the effect of fibrillar Aβ42 on forskolin (Fsk)-induced syncytialization by analyzing the secretion and induction of human chorionic gonadotropin β-subunit (β-hCG) (Wice et al, 1990; Gerbaud & Pidoux, 2015) and induction of syncytin-1 expression (Mi et al, 2000) in BeWo cells. Here, pretreatment of BeWo cells with Aβ42 fibrils in a micromolar range significantly reduced the secretion and induction of β-hCG by 40% in the medium and reduced the expression of syncytin-1 by 65% (Fig 4A). We observed that nonaggregated Aβ did not affect the secretion of β-hCG (Fig S6), which corroborated the detrimental effects of aggregated Aβ on CTB syncytialization.

Cell–cell adhesion proteins such as ZO-1 and E-cadherin are required for CTB syncytialization (Pidoux et al, 2010; Iwahashi et al, 2019). Aβ aggregates disrupt membrane localization of tight junction proteins, at least in part, by inducing excess autophagy (Marco & Skaper, 2006; Chan et al, 2018). We hypothesized that Aβ fibrils might also disrupt the membrane localization of ZO-1 and E-cadherin in cytotrophoblasts. Therefore, we next investigated the effect of Aβ42 fibrils on the subcellular localization of ZO-1 and E-cadherin in BeWo cells and how this affects CTB syncytialization. Immunofluorescence analysis with an anti-ZO-1 antibody and an anti-E-cadherin antibody revealed that Aβ42 fibrils disrupted the membrane localization of ZO-1 and E-cadherin (Fig 4B, arrowheads). Immunoblots showed significant reductions in ZO-1 and E-cadherin protein levels in Aβ42 fibril-treated BeWo cells (34% for ZO-1 and 45% for E-cadherin, Fig 4C). A previous study suggested that Aβ fibril treatment enhanced autophagy and thereby reduced levels of cell adhesion–related proteins including ZO-1 in endothelial cells (Chan et al, 2018). Turnover of E-cadherin is at least partly regulated via the autophagic pathway (Santarosa & Maestro, 2021). Here, Aβ42 fibril–treated BeWo cells showed a significant 260% increase in LC3 levels, which suggested that increased autophagy resulted in decreases in and loss of membrane localization of ZO-1 and E-cadherin (Fig 4C). Although Aβ42 fibrils increased the mRNA expression of E-cadherin (Fig S7), these results suggest that Aβ42 fibrils interfered with Fsk-induced syncytialization by disrupting the proper membrane localization of cell adhesion–related proteins that are required for syncytialization. We also excluded the possibility that Aβ fibrils induced cell death in BeWo cells (Fig S8).

### Aβ fibrils inhibited syncytialization of human primary cultured CTBs

Because BeWo cells require Fsk for syncytialization and human CTBs spontaneously undergo syncytialization without Fsk (Costa, 2016), we further investigated the effect of Aβ fibrils on human placentation using primary cultured human trophoblast cells. We isolated trophoblast cell fractions from human normal placentas as previously described (Simon et al, 2017). These cells secreted β-hCG 72 h after seeding, which suggested that CTBs were successfully isolated (Fig 5A). We also confirmed by means of Western blotting that human primary cultured CTBs produced Aβ40/42. We pretreated human CTBs with Aβ42 fibrils (10 μM) for 24 h, after which the culture media were replaced with fresh media containing Aβ42 fibrils (10 μM), and incubation continued for an

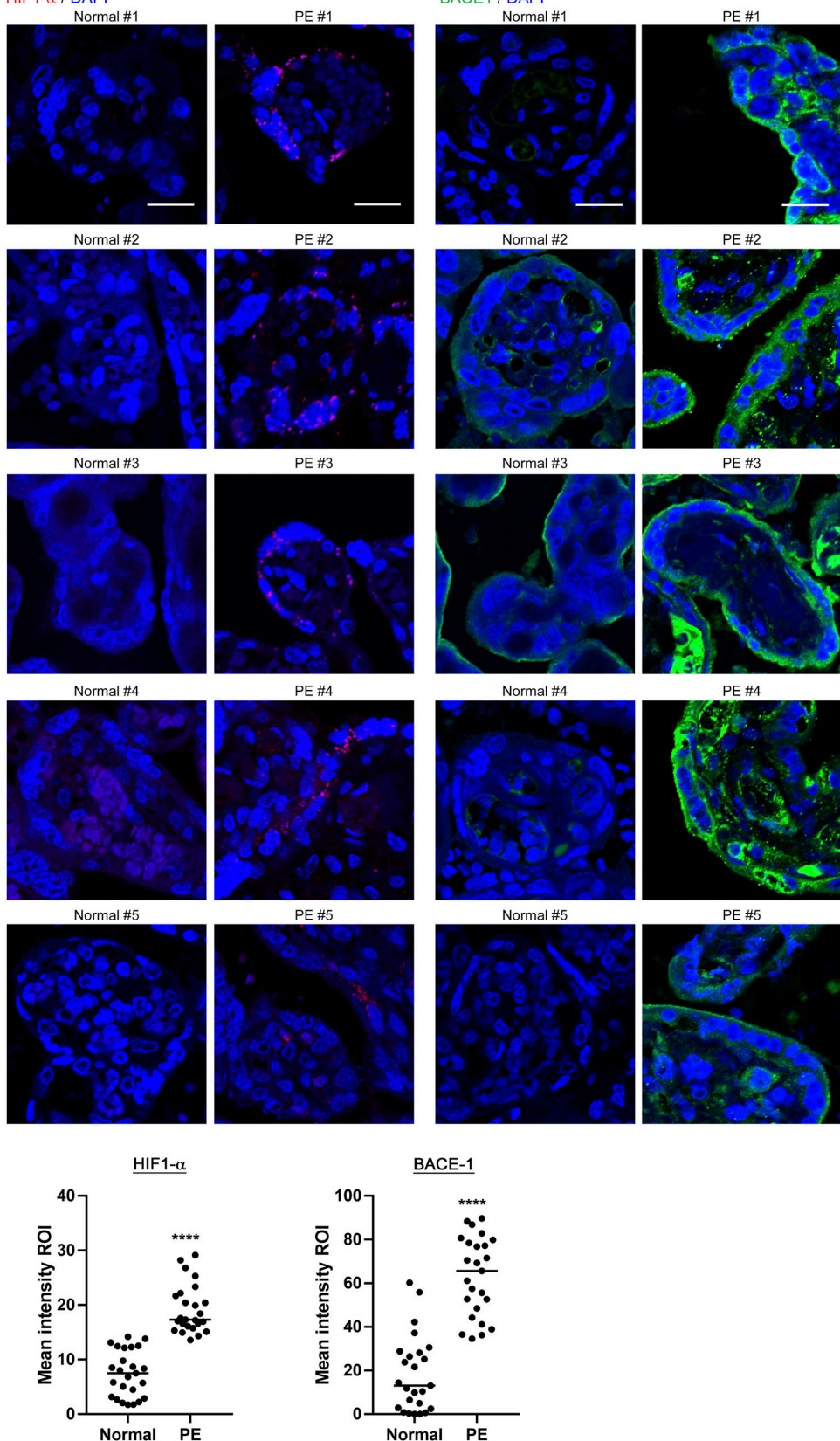

**Figure 2. HIF1-α and BACE1 levels in PE placentas.**
Sections were immunostained with the anti-HIF-1α antibody or the anti-BACE1 antibody. Nuclei were counterstained with DAPI. Graphs show the quantification of the mean intensities of HIF1-α and BACE-1 signals. ****P < 0.0001. Scale bars, 20 μm.
Source data are available for this figure.

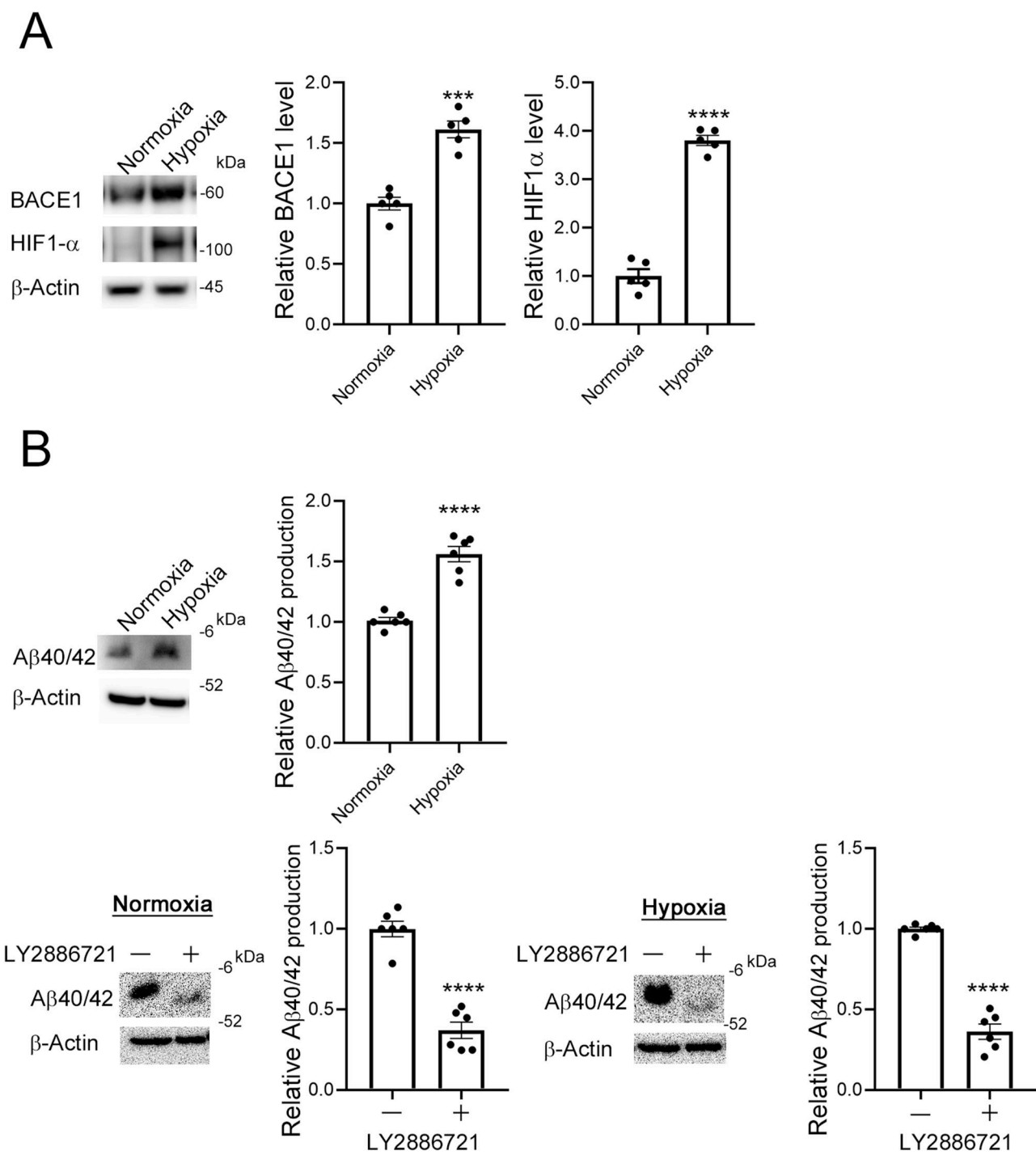

**Figure 3. Hypoxia increased the level of BACE1 and Aβ production in BeWo cells.**
**(A)** BeWo cells were cultured under normoxic conditions (20% O$_2$) or hypoxic conditions (2% O$_2$) for 3 h, after which protein levels of HIF-1α and BACE1 were analyzed by means of Western blotting with the anti-HIF-1α antibody and the anti-human BACE1 antibody, respectively. β-Actin served as the loading control. Graphs show quantification of HIF-1α and BACE1. Data are means ± SEM ($n$ = 5). ***$P$ < 0.001; ****$P$ < 0.0001. **(B)** Hypoxia promoted Aβ production in BeWo cells. Aβ40/42 generated by BeWo cells was analyzed by means of Western blotting with the anti-Aβ (Nt) β001 antibody. BeWo cells were cultured in Opti-MEM containing 2% FBS with or without LY2886721 for 24 h under normoxic conditions (20% O$_2$) or hypoxic conditions (2% O$_2$). Generated Aβ40/42 was analyzed. β-Actin served as the loading control. Graphs show quantification of Aβ40/42. Data are means ± SEM ($n$ = 6). ***$P$ < 0.001; ****$P$ < 0.0001.
Source data are available for this figure.

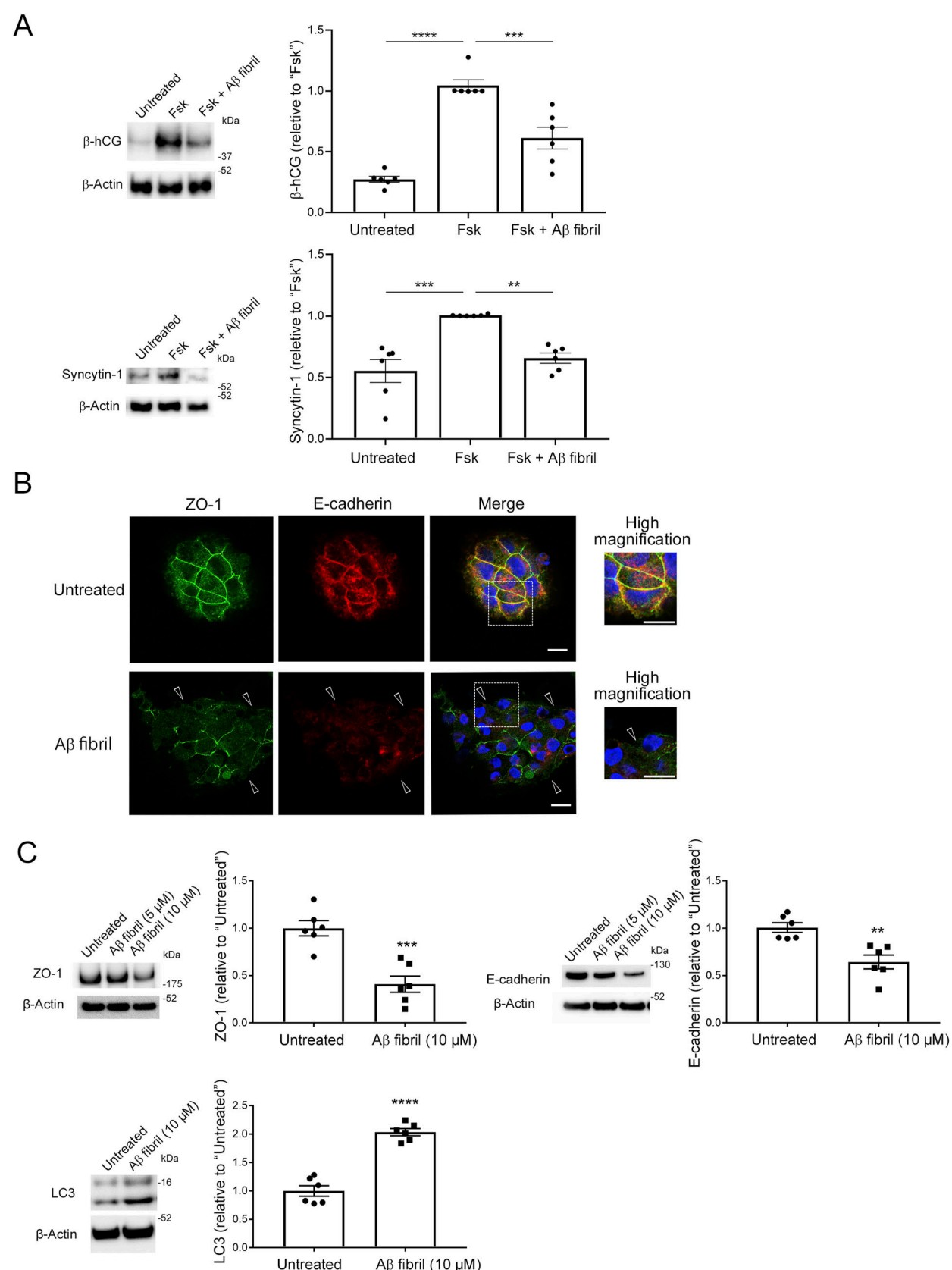

additional 48 h. We then analyzed the induction and secretion of β-hCG and induction of syncytin-1 by means of Western blotting, which revealed that treatment of human CTBs with Aβ42 fibrils significantly reduced β-hCG induction and secretion—38% and 65%, respectively (Fig 5B). Aβ42 fibrils also reduced syncytin-1 induction by 65%, which was similar to the findings for BeWo cells (Fig 5B). We then asked whether Aβ42 fibrils would affect subcellular localization of ZO-1 and E-cadherin in human CTBs. Immunocytochemical analysis of Aβ42 fibril–treated human trophoblasts revealed disrupted membrane localization of ZO-1 and E-cadherin (Fig 5C, arrowheads). These results strongly support the finding that Aβ fibrils disrupted the proper localization of ZO-1 and E-cadherin at the cell–cell border and thereby inhibited syncytialization of CTBs.

## Discussion

PE is a complicated syndrome with multifactorial pathology, whose etiology is not well understood. Given that the current definitive treatment of PE is termination of pregnancy, elucidating the mechanisms of placentation and placental defects is important for prevention of PE and development of novel PE therapies. Numerous studies have suggested that the aberrant accumulation of misfolded proteins and their aggregates underlies the pathology of human diseases such as AD, Parkinson's disease, age-related macular degeneration, arthritis, and p53-mutant cancers (Chiti & Dobson, 2017; Iwahashi et al, 2020). Other recent studies have demonstrated the accumulation of misfolded proteins in PE placentas, suggesting that PE also belongs to a class of protein conformational diseases (Cater et al, 2019; Cheng et al, 2022). Several studies have reported placental deposition of aggregates of Aβ and transthyretin, whose mutants cause the most common form of hereditary amyloidosis (Raz et al, 1970; Buxbaum et al, 2008), in PE placentas (Cater et al, 2019; Cheng et al, 2022). However, these studies did not address how these aggregates contribute to poor placentation. Down syndrome is caused by trisomy of chromosome 21, on which APP is located, and is characterized by marked accumulation of Aβ fibrils in the brain (Burger & Vogel, 1973; Wisniewski et al, 1985; Oyama et al, 1994). The study by Wong et al reported that overexpression of APP in BeWo cells inhibited syncytialization (Wong et al, 2018), but the effect of Aβ fibrils on syncytialization of BeWo cells was not investigated. Our study provides the evidence and mechanism that Aβ fibrils indeed inhibit STB formation and may contribute to the PE pathology. Aβ

aggregates reportedly enhanced autophagy in HTR8/SVneo cells, and an excessive autophagy may exacerbate PE (Gao et al, 2015, 2024). On the other hand, immediately after the implantation, an embryo has to survive a severe low oxygen tension condition because of the lack of vasculature and oxygen supply (Rodesch et al, 1992; Pollheimer et al, 2018). Thus, EVTs aggressively invade the uterine decidua and myometrium to remodel the maternal spiral arteries and develop a vasculature for the maternal–fetal interface (Pijnenborg et al, 1980; Tayade et al, 2006). Here, we also found that HTR8/SVneo cells, a widely accepted model of EVTs (Graham et al, 1993), produced much less amount of Aβ than BeWo cells and nM range of Aβ promoted EVT invasiveness (Fig S9A). We identified 1,444 differentially expressed genes in Aβ-treated HTR8/SVneo cells (Fig S9B), and our transcriptome analysis suggested that quite low concentration of Aβ enhanced EVT invasiveness by activating R-HSA-1442490 (collagen degradation) and R-HSA-1592389 (activation of matrix metalloproteases) (Fig S9C). These results suggest that although Aβ fibrils are detrimental to the placenta, nM range of Aβ monomers may have a physiological function in early pregnancy. Elucidation of the effects of Aβ or Aβ fibrils in the endovascular crosstalk between trophoblasts and vascular endothelial cells is a future research topic. Currently, the production of Aβ across gestation has not been studied, and the stage at which Aβ begins to deposit in PE placentas is unclear. Additional studies are necessary to elucidate the significance of Aβ metabolism and deposition in the physiology and pathology of the human placenta.

Because placental hypoxia plays an important role in the pathophysiology of PE (Tong et al, 2022), we hypothesized that hypoxia in the PE placenta increases Aβ production and that Aβ deposited in the placenta affects syncytialization Here, pretreatment of BeWo cells and human CTBs with Aβ42 fibrils significantly reduced β-hCG secretion and induction in the medium and of syncytin-1 expression, indicating that Aβ42 fibrils inhibited syncytialization and STB formation by CTBs. Knockdown of ZO-1 decreased cell–cell fusion and subsequent trophoblastic differentiation (Pidoux et al, 2010), and reduced cell surface expression of E-cadherin led to dysfunctional cell–cell adhesion and disturbed syncytialization (Iwahashi et al, 2019). The proper function of cell–cell adhesion proteins depends on their appropriate membrane localization, but is not reflected in lower protein levels (Tsukita et al, 2001; Yap & Kovacs, 2003). Aβ42 assemblies disrupted the membrane localization of tight junctional proteins, including ZO-1, by inducing autophagy in murine cerebral capillary endothelial cells (Kook et al, 2012), and E-cadherin turnover was regulated via the endocytosis and autophagic pathway (Santarosa &

---

**Figure 4. Aβ42 fibrils inhibited syncytialization of BeWo cells by inducing loss of membrane localization of cell–cell adhesion proteins.**
**(A)** BeWo cells were pretreated with Aβ42 fibrils (10 μM) in serum-free Opti-MEM for 12 h and then were stimulated with Fsk (50 μM) for 48 h. The effect of Aβ fibrils on syncytialization of BeWo cells was analyzed by means of Western blotting with the anti-hCG β antibody and the anti-syncytin-1 antibody. β-Actin served as the loading control. Graphs show quantification of β-hCG and syncytin-1. Data are means ± SEM (n = 6). **P < 0.01; ***P < 0.001; ****P < 0.0001. **(B)** BeWo cells were cultured on cover glasses and treated with Aβ42 fibrils (10 μM) for 24 h, after which they were fixed in 4% PFA and stained with the anti-ZO-1 antibody or the anti-E-cadherin antibody. Arrowheads indicate loss of membrane localization of ZO-1 and E-cadherin in Aβ fibril–treated BeWo cells. Nuclei were counterstained with DAPI. Scale bars, 20 μm. **(C)** BeWo cells were treated with Aβ42 fibrils (10 μM) in serum-free Opti-MEM for 24 h. Quantitative analysis of ZO-1 and E-cadherin of Aβ42 fibril-treated BeWo cells was performed using Western blotting with the anti-ZO-1 antibody and the anti-E-cadherin antibody. Evaluation of the effect of Aβ42 fibrils on autophagy of BeWo cells was performed using Western blotting with an anti-LC3 antibody. β-Actin served as the loading control. Graphs show quantification of ZO-1, E-cadherin, and LC3. Data are means ± SEM (n = 6). **P < 0.01; ***P < 0.001; ****P < 0.0001.
Source data are available for this figure.

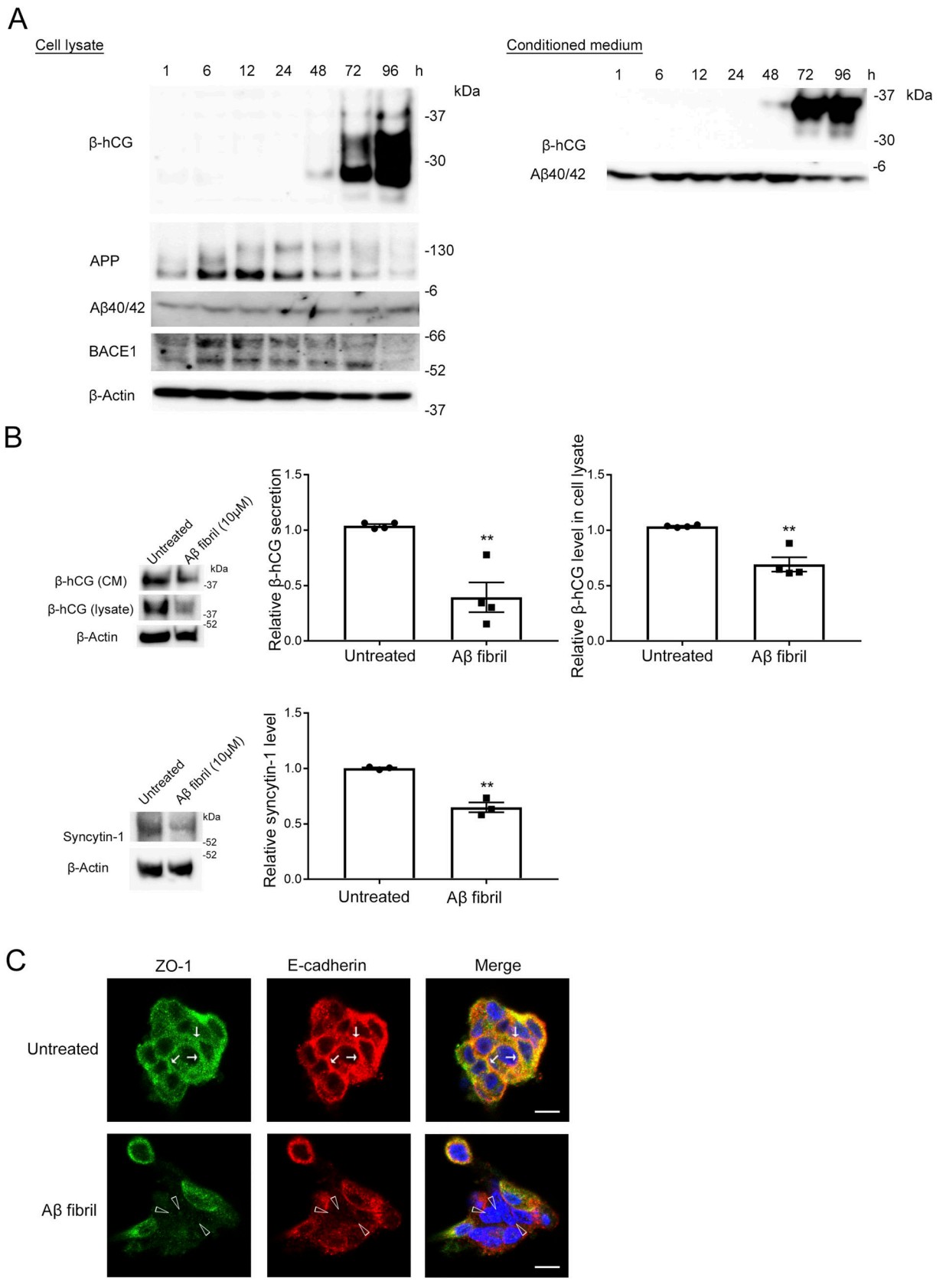

Maestro, 2021). Therefore, BeWo cells treated with Aβ42 fibrils showed a significant increase in LC3 levels, suggesting that increased autophagy led to reduced ZO-1 and E-cadherin. Pretreatment of CTBs with Aβ42 fibrils induced loss of membrane localization of cell–cell adhesion proteins, which are essential for CTB syncytialization. These results suggest that Aβ deposition is increased in the hypoxic PE placenta and that Aβ fibrils enhance autophagy of CTB, thereby disrupting the proper membrane localization of cell adhesion–related proteins such as ZO-1 and E-cadherin and inhibiting syncytialization, which may be a cause of placental dysfunction and PE. Because the loss of membranous E-cadherin was found to slow the fusion process in BeWo cells (Iwahashi et al, 2019), it is possible that Aβ fibrils also slow the CTB fusion process.

Our results suggest that Aβ productions by CTBs are at least partially regulated by hypoxia and HIF1-α; however, other factors that were implicated in increased Aβ production, such as endoplasmic reticulum stress (Lian et al, 2011; Fu et al, 2015; Jung et al, 2020) and oxidative stress (Li et al, 2004; Raijmakers et al, 2004), may be involved in Aβ production by CTBs in the late stage of gestation. It is notable that endoplasmic reticulum stress and oxidative stress were reportedly induced in PE placentas (Raijmakers et al, 2004; Veerbeek et al, 2015). Further investigation is necessary to identify the stress involved in Aβ deposition in PE placentas. The stage at which Aβ begins to deposit in PE placentas is still unclear. Defective spinal artery remodeling can result in chronic and pathogenic hypoxia, which would lead to an increase in Aβ42 production and Aβ deposition. Additional studies are needed to elucidate the significance of Aβ metabolism and deposition in the pathology of the human placenta. Cheng et al reported that hypoxia and subsequent induction of endoplasmic reticulum stress induced abnormal accumulation of TTR aggregates in the placental junctional zone (Cheng et al, 2022); however, the contribution of TTR aggregates to poor placentation was not fully clarified. Although our results clearly indicate the detrimental role of Aβ fibrils in the maintaining pregnancy, future studies are also needed to elucidate the pathological roles of different protein aggregates. To date, no reports have been found showing Aβ deposition in placentas other than those from PE. Although the deposition of protein aggregates, including those of Aβ and TTR, has been reported in PE, the presence and role of these protein deposits in the placenta under physiological or pathological conditions remained to be elucidated. We found that Aβ is produced by CTBs and EVTs in a basal state. The amount of Aβ is determined by the balance between the production and the clearance (Selkoe & Hardy, 2016). A sustained imbalance in the production and clearance of Aβ leads to its aggregation and deposition. Although nonaggregated Aβ did not affect the

syncytialization in BeWo cells, nM or pM Aβ promoted EVT invasion. This suggests that Aβ has a physiological function in the placenta. On the other hand, Aβ fibrils were detrimental to CTBs and EVTs. Thus, normal placentas may express BACE1 and produce small amounts of Aβ. Our results suggest that chronic hypoxia in PE placentas results in increased BACE1 expression and sustained Aβ production. Eventually, this leads to Aβ aggregation and deposition because the aggregation process depends on the local Aβ concentration. Elucidating the physiological roles of Aβ in CTB or STB functions deserves future investigation. The aggregation processes of proteins, including those of Aβ42, are concentration-dependent and require concentrations higher than the micromolar range (Burdick et al, 1992; Knowles et al, 2014). Because amyloid fibrils and Aβ are in equilibrium, Aβ concentrations around amyloid deposits are expected to be quite high (DeMattos et al, 2002). Therefore, we propose that chronic hypoxia in PE leads to an elevated local concentration of Aβ through a sustained increase in its production. This ultimately results in Aβ fibrillogenesis and the deposition of Aβ fibrils. In addition, several biomolecules such as sulfated glycosaminoglycans are known to be critical for fibrillogenesis of amyloidogenic proteins (Nishitsuji & Uchimura, 2017). Accordingly, Aβ co-deposited with highly sulfated heparan sulfate (Fig S1), which supports the involvement of sulfated glycosaminoglycans in placenta Aβ deposition. Further investigation of this topic is another future challenge.

We showed here that Aβ fibrils are detrimental to CTB syncytialization in PE placentas. It further remains to be analyzed whether Aβ deposition in the placenta directly contributes to the pathogenesis of PE. The limitation of this study is that we used a cellular model for demonstrating hypoxia-enhanced Aβ production. Lai et al reported a preclinical PE model by coupling low oxygen tension and an interleukin-10 deficiency (Lai et al, 2011). An important future research topic will be the use of preclinical models to further investigate the role of hypoxia in placental Aβ production and the role of Aβ aggregates in the etiology and pathology of PE.

## Materials and Methods

### Materials

Aβ peptides (human, 1–42) were purchased from Peptide Institute (Osaka, Japan). Rabbit polyclonal anti-β-actin and rabbit polyclonal anti-microtubule–associated protein 1A/1B-light chain 3 (LC3) antibodies were purchased from Medical and Biological Laboratories. Rabbit monoclonal anti-HIF1α antibody was purchased from Cell Signaling Technology, and a mouse monoclonal

**Figure 5. Aβ42 fibrils inhibited primary cultured human CTB syncytialization.**
**(A)** Primary cultured human CTBs were isolated from third-trimester human placentas and cultured in trophoblast medium supplemented with 5% FBS for 96 h, after which the expression of APP and BACE1 and the production and secretion of β-hCG and Aβ40/42 were analyzed by means of Western blotting. **(B)** Primary cultured human CTBs were treated with Aβ1-42 fibrils (10 μM) in serum-free Opti-MEM for 72 h. The effect of Aβ fibrils on syncytialization of human CTBs was analyzed by means of Western blotting with the anti-hCG β antibody and the anti-syncytin-1 antibody. β-Actin served as the loading control. Graphs show quantification of β-hCG and syncytin-1. Data are means ± SEM ($n$ = 3 or 4). **$P$ < 0.01. **(C)** Primary cultured human CTBs were cultured on cover glasses and were treated with Aβ1-42 fibrils (10 μM) for 18 h, after which they were fixed in 4% PFA. They were then stained with the anti-ZO-1 antibody or the anti-E-cadherin antibody. Arrows indicate ZO-1 and E-cadherin located at the cell–cell border, and arrowheads indicate loss of membrane localization of these proteins. Nuclei were counterstained with DAPI. Scale bars, 20 μm. Source data are available for this figure.

anti-BACE1 antibody was from R&D Systems. Rabbit polyclonal anti-$\beta$-hCG antibody was obtained from Proteintech. Rabbit polyclonal anti-ERVWE1/HERV/Syncytin antibody, a rabbit polyclonal anti-ZO-1 tight junction protein antibody, and a mouse monoclonal anti-E-cadherin antibody were purchased from Lifespan Biosciences, Abcam, and BD Biosciences, respectively. LY2886721, the BACE1 inhibitor, was purchased from Abcam. Roxadustat was purchased from Selleck Chemicals. Anti-human BACE1 (C) rabbit IgG affinity-purified antibody was purchased from Immuno-Biological Laboratories. The 22C11 anti-APP antibody was from Thermo Fisher Scientific. $\beta$-Secretase-cleaved Nt-specific rabbit polyclonal anti-A$\beta$ antibody ($\beta$001) was established as previously described (Lippa et al, 1999). RB4CD12 anti-S-domain antibody was kindly provided by Dr. Toin H. van Kuppevelt (Nijmegen Center for Molecular Life Sciences, Radboud University Nijmegen Medical Center, Nijmegen, The Netherlands) (Dennissen et al, 2002).

## Human tissue collection

This study was approved by the Ethics Committee of Wakayama Medical University (authorization no. 1690) and was conducted according to the tenets of the Declaration of Helsinki. All patients gave written informed consent for the use of tissue samples in this study. Third-trimester human placentas were collected after vaginal deliveries or cesarean sections. HDP (hypertensive disorders of pregnancy) is defined as hypertension (blood pressure ≥ 140/90 mmHg) during the pregnancy. PE is one type of HDP and is accompanied by one or more of the following new-onset conditions at or after 20 wk of gestation: proteinuria (≥300 mg protein/ 24 h); other maternal organ dysfunctions including liver involvement without any underlying diseases; progressive kidney damage; stroke; neurological complications such as clonus, eclampsia, visual field disturbance, and severe headache except for primary headache; hematological complications such as thrombocytopenia caused by an HDP-related platelet count lower than 150,000/$\mu$l, disseminated intravascular coagulation, and hemolysis; and uteroplacental dysfunction such as fetal growth restriction, abnormal umbilical artery Doppler waveform results, and stillbirth. All the above-mentioned symptoms and signs become normal by 12 wk postpartum. The eligibility of the PE cases was determined according to the diagnostic criteria of the International Society for the Study of Hypertension in Pregnancy (Brown et al, 2018). Cases involving multiple pregnancies, fetal chromosomal abnormalities, and fetal anomalies were excluded. For immunohistochemical analysis, placental tissue samples from PE patients and gestational age-matched controls were washed in ice-cold PBS (pH 7.2) before fixation with 4% PFA in PBS ($n$ = 5 for normal pregnancy and $n$ = 5 for PE).

## Immunohistochemistry

Paraffin-embedded placenta blocks were cut into 3-$\mu$m-thick sections, after which they were deparaffinized and rehydrated. Epitopes were then retrieved via heat by boiling sections in a pressure cooker in citrate-based Antigen Unmasking Solution (pH 6.0; Vector Laboratories) for 20 min. Sections were then incubated with the $\beta$001 anti-A$\beta$ antibody (1:1,000), or an anti-HIF-1$\alpha$ antibody (1:100) and an anti-BACE1 antibody (1:50), followed by the use of an Alexa Fluor 488–conjugated polyclonal goat anti-mouse IgG (1:300; Thermo Fisher Scientific), a Cy3-conjugated polyclonal goat anti-rabbit IgG, or an Alexa Fluor 488–conjugated polyclonal goat anti-mouse IgG (1:400; Thermo Fisher Scientific). For detection of A$\beta$ aggregates, sections were washed with PBS, incubated in ProteoStat solution (1:2,000 in the ProteoStat assay buffer) for 3 min, and then destained in 1% acetic acid for 20 min at RT (Matafora et al, 2020). Endogenous autofluorescence derived from red blood cells was quenched using the TrueVIEW Autofluorescence Quenching Kit (Vector Laboratories), and the sections were mounted with Vectashield mounting medium with DAPI (Vector Laboratories). Specimens were then examined with an LSM700 confocal microscope (Plan-Apochromat, 20×/0.8; Carl Zeiss). For quantification, five regions of interest (ROIs; 16 $\mu$m × 16 $\mu$m) were set on villi in each placenta tissue, and mean intensities were determined using ZEN 3 blue edition (Carl Zeiss). $\beta$001- and ProteoStat-positive areas were determined using the Coloc. Tools of ZEN 3. Sections were also immunostained with the $\beta$001 rabbit polyclonal anti-amyloid $\beta$ antibody and RB4CD12 (1:100), a marker of co-deposition of amyloid/protein aggregates. Cy3-conjugated monoclonal anti-vesicular stomatitis virus G glycoprotein (1:300; Sigma-Aldrich) was used for the secondary antibody for RB4CD12. $\beta$001- and RB4CD12-positive areas were determined using the Coloc. Tools of ZEN 3.

## Cell culture

Human choriocarcinoma-derived BeWo cells were purchased from the American Type Culture Collection, and they were authenticated by JCRB Cell Bank (report no. KBN0410; National Institute of Biomedical Innovation, Japan). We successfully analyzed syncytialization using them (Yamamoto et al, 2017; Iwahashi et al, 2019, 2021). BeWo cells were maintained in minimal essential medium-$\alpha$ (MEM-$\alpha$) (Wako Pure Chemicals) supplemented with 15% FBS and the antibiotic mixture. Cells were cultured at 37°C in an atmosphere of 5% $CO_2$ and 95% air.

## Hypoxic treatments

Hypoxic conditions (2% $O_2$) were established using the BIONIX-1 hypoxic culture kit (Sugiyama-Gen) containing an AnaeroPack-Anaero 5% system (an oxygen absorber; Mitsubishi Gas Chemical), an OXY-1 oxygen monitor (JIKCO), an AnaeroPouch (Mitsubishi Gas Chemical), and plastic clips for sealing pouches. Cells were grown in 6-cm dishes or 6-well, 12-well, or 24-well plates, and an OXY-1 oxygen monitor and an oxygen absorber were arranged in a pouch, and the left open side was sealed with a clip. When the $O_2$ concentration in the pouch reached 2%, the pouch between the culture dish or plate and the oxygen absorber was sealed with another clip to prevent additional oxygen absorption, and the pouch was maintained at 37°C. The cells were then cultured for an additional 3 or 24 h in hypoxic conditions. HIF1-$\alpha$ and BACE1 protein levels were analyzed by Western blotting using an anti-HIF-1$\alpha$ antibody (1:1,000) and an anti-human BACE1 (C) antibody (1:50), respectively, as described below.

### Aβ fibril preparation

Aβ fibrils were prepared as previously described (Zhang et al, 2017). Briefly, chemically synthesized Aβ peptide (human, 1–42; Peptide Institute) was dissolved in 0.1% NH₃ to prepare a 1 mM stock solution. Stock solution was diluted in PBS to 200 $\mu$M and was incubated at 37°C for 5 d to prepare Aβ fibrils. The formation of Aβ fibrils was confirmed by measuring the fluorescence intensity of thioflavin T and by transmission electron microscopy analysis (Fig S10A and B). We also determined the fibril content to be ~94% through analysis with native-PAGE, followed by Western blotting with β001 and dot blotting with the OC anti-amyloid antibody (Kayed et al, 2007) (Fig S10C).

### Western blot analysis

Aβ generated by BeWo cells was analyzed by means of Western blotting. BeWo cells were cultured in Opti-MEM with 2% FBS with or without the BACE1 inhibitor (LY2886721) for 24 h under normoxic conditions (20% O₂) or hypoxic conditions (2% O₂), respectively. CM samples were then collected and centrifuged at 2,000$g$ for 30 min to remove debris, after which 100% trichloroacetic acid (TCA) in PBS was added to obtain a final concentration of 10%, and the mixture was incubated for 30 min at 4°C, followed by centrifugation at 12,000$g$ for 10 min at 4°C. Precipitates were dissolved in UTB (9 M urea, 2% Triton X-100, 5% 2-mercaptoethanol) and were sonicated on ice. The 2× sodium dodecyl sulfate–polyacrylamide gel electrophoresis (SDS–PAGE) sample buffer (4% sodium dodecyl sulfate, 12% 2-mercaptoethanol, 0.1 M Tris, 20% glycerol, and 0.01% bromophenol blue) was then added to the samples, which were sonicated on ice to prepare PAGE samples that were subjected to Western blotting as described below.

The effects of Aβ fibrils on CTB differentiation were investigated by analyzing β-hCG production and syncytin-1 expression via Western blotting. Fsk was used as an inducer of syncytialization of BeWo cells. BeWo cells were pretreated with Aβ42 fibrils (10 $\mu$M) in serum-free Opti-MEM for 12 h and then stimulated with Fsk (50 $\mu$M) for 48 h. To analyze ZO-1, E-cadherin, and LC3 expression, BeWo cells were cultured in serum-free Opti-MEM in the presence or absence of Aβ42 fibrils (10 $\mu$M) for 24 h. To investigate β-hCG production and secretion, CM samples were collected and centrifuged at 2,000$g$ for 30 min to remove debris. CM obtained was mixed with 2× SDS–PAGE sample buffer and heated at 95°C for 5 min. To determine protein levels of β-hCG, syncytin-1, ZO-1, E-cadherin, and LC3, cells were fixed with 10% TCA and whole-cell lysates were prepared as described above. Samples were then subjected to SDS–PAGE with 5–20% gels (Wako Pure Chemicals) and were transferred to polyvinylidene difluoride membranes (Millipore). To analyze Aβ production, PAGE samples were subjected to NuPAGE with NuPAGE MES SDS Running Buffer and NuPAGE 4–12%, Bis-Tris gels (Thermo Fisher Scientific). Membranes were blocked in the EzBlock Chemi blocking solution (ATTO) or 5% BSA (Proliant Biologicals) and 0.1% Tween-20 in Tris-buffered saline at RT for 1 h, after which they were incubated with the anti-hCG β antibody (1:1,000), the anti-syncytin-1 antibody (1:1,000), the anti-ZO-1 antibody (1:1,000), the anti-E-cadherin antibody (1:1,000), or the anti-LC3 antibody (1:1,000), followed by a preabsorbed horseradish peroxidase–conjugated anti-rabbit or anti-mouse IgG (1:10,000, Jackson ImmunoResearch Laboratories). Signals were visualized using ImmunoStar LD Chemiluminescence Reagent (Wako Pure Chemicals) and an Amersham ImageQuant 800 system (Cytiva). To measure Aβ, membranes were incubated with the β001 rabbit polyclonal anti-Aβ (Nt) antibody (1:10,000) followed by a preabsorbed horseradish peroxidase–conjugated anti-rabbit (1:10,000; Jackson Immuno-Research Laboratories).

### Immunocytochemistry

BeWo cells were cultured on cover glasses and treated with Aβ fibrils (10 $\mu$M) for 24 h at 37°C, after which they were fixed in 4% PFA in PBS at RT for 20 min. After the cells were washed three times with PBS, they were blocked and permeabilized with 20% Animal-Free Blocker (Vector Laboratories) containing 0.05% saponin (Wako Pure Chemicals) in PBS for 20 min at RT. The samples were then incubated with the anti-ZO-1 antibody (1:100) or the anti-E-cadherin antibody (1:200) followed by incubation with Alexa Fluor 488–conjugated polyclonal goat anti-rabbit IgG (1:300) or Alexa Fluor 568–conjugated polyclonal goat anti-mouse IgG (1:300). Specimens were mounted with the Vectashield mounting medium with DAPI and were examined with an LSM700 confocal microscope (C-Apochromat 40×/1.2 for BeWo cells; C-Apochromat 63×/1.2 for primary CTBs).

### Isolation of human trophoblasts

All patients provided written informed consent for the use of tissue samples. Villous CTBs were isolated as previously described (Simon et al, 2017). Placental tissues from four normal full-term pregnancies were obtained by elective cesarean section before the onset of labor. Villous tissues were washed three times with PBS, and chorionic and basal plates and large vessels were removed, after which tissues were again washed with PBS and minced. We digested villous tissues with trypsin (0.25%; Thermo Fisher Scientific) and DNase (0.03 mg/ml; Merck), and the supernatants obtained were filtered using 100-$\mu$m cell strainers (SPL Life Sciences). FBS samples (5 ml) were added slowly to the bottoms of the tubes containing the supernatants, which were centrifuged at 1,250$g$ for 15 min at RT. After we gently removed the supernatants, we resuspended cell pellets that contained red blood cells and CTBs in RPMI 1640 medium supplemented with 10% FBS. Suspensions were layered on top of a 20–60% Percoll (Cytiva) gradient in Hanks' Balanced Salt Solution (Thermo Fisher Scientific). After centrifugation at 1,250$g$ for 20 min at RT, the CTB layers were collected. RPMI 1640 medium supplemented with 10% FBS was added to the CTB layers and subjected to additional centrifugation at 1,250$g$ for 15 min at RT. Isolated primary human CTBs were seeded at a density of 1.0 × 10⁶/4 wells in a 12-well plate and grown in trophoblast medium (ScienCell Research Laboratories) supplemented with 5% FBS and an antibiotic mixture containing penicillin and streptomycin (Life Technologies). Expression of APP, expression of BACE1, and Aβ production in human primary cultured

CTBs were confirmed using Western blotting with the 22C11 anti-APP antibody, the BACE1 antibody, and the β001 antibody, respectively. The effects of Aβ fibrils on CTB differentiation were analyzed by means of Western blotting with the anti-β-hCG antibody and the anti-syncytin-1 antibody as described above except that primary CTBs were pretreated with Aβ fibrils (10 μM in Opti-MEM) for 24 h and then incubated in fresh Opti-MEM with 10 μM Aβ fibrils for an additional 48 h. The effects of Aβ fibrils on the expression and subcellular localization of ZO-1 and E-cadherin were immunocytochemically analyzed as described above, except that primary CTBs were cultured on poly-L-lysine–coated cover glasses for 9 h and treated with Aβ fibrils (10 μM) for 18 h.

### Thioflavin T (ThT) fluorescence assay and transmission electron microscopy (TEM)

ThT (10 μM) in an Aβ42 fibril solution (200 μM in PBS, pH 7.4) was exited at 445 nm, and the fluorescence intensity was recorded from 470 to 600 nm using a MTP-900Lab microplate reader (Hitachi High-Tech Science). For TEM analysis, Aβ42 fibrils (200 μM in PBS) were spread on carbon film–coated copper grids and negatively stained twice with 2% uranyl acetate for 1 min. The grids were then examined under a JEM-1400Plus transmission electron microscope (JEOL) with an acceleration voltage of 100 kV. Digital images (3,296 × 2,472 pixels) were obtained using a charge-coupled device camera (EM-14830 RUBY2; JEOL).

### Analysis of E-cadherin mRNA

To analyze the expression of E-cadherin, BeWo cells were cultured in serum-free Opti-MEM in the presence or absence of Aβ42 fibrils (10 μM) for 24 h. Total RNA was obtained using the TRIzol reagent (Thermo Fisher Scientific). RT–qPCR was carried out with the CFX96 Touch Real-Time system (Bio-Rad Laboratories) and the iTaq Universal SYBR Green one-step kit (Bio-Rad). RT–qPCR experiments were performed in triplicate. Data were processed using the Bio-Rad CFX Manager version 3.1 (Bio-Rad Laboratories), and expression levels were calculated via the comparative ΔΔCt method using *GAPDH* as the reference gene. The sequences of the primers used were as follows: E-cadherin, forward: CAAATCCAACAAAGACAAAGAAGGCAA, reverse: ATGACAGACCCCTTAAAGACCTCCT; *GAPDH*, forward: GAGTCAACGGATTTGGTCGT, reverse: GACAAGCTTCCCGTTCTCAG.

### ELISA

BeWo cells were cultured in Opti-MEM with 2% FBS with or without the LY2886721 BACE inhibitor for 24 h under normoxic conditions (20% $O_2$) or hypoxic conditions (2% $O_2$). Conditioned medium (CM) samples were harvested, and Aβ levels in these samples were quantified by means of the human β-amyloid (1–40) ELISA kit (Wako Pure Chemicals) and an MTP-320Lab microplate reader (Corona Electric) according to the manufacturers' instructions. Concentrations of Aβ generated by BeWo cells were determined on the basis of a standard curve that was obtained using the standard solution.

### Analysis of the effects of HIF1-α stabilization on BACE1 expression

For analysis of the effects of HIF1-α stabilization on BACE1 expression, we used roxadustat (Selleck Chemicals), which inhibits prolyl hydroxylase and stabilizes HIF, as an HIF1-α stabilizer. BeWo cells were treated with roxadustat (0, 5, 10, or 25 μM) in MEM-α medium containing 15% FBS for 16 6 h, after which whole-cell lysates were prepared and subjected to Western blotting with the anti-HIF-1α antibody (1:1,000) and the anti-human BACE1 (C) antibody (1:50), respectively, as described in the Materials and Methods section.

### Matrigel invasion assay

HTR8/SVneo cells were used as an extravillous trophoblast (EVT) model. EVT invasion was analyzed using Boyden chamber plates with Matrigel-coated 24-well Transwell inserts. Inserts were coated with 50 μl of 1 mg/ml Matrigel matrix (BD Biosciences). Aβ40 (0.01–100 nM; Peptide Institute) or BACE1 inhibitor (LY2886721, 80 nM; Abcam) was added to the Matrigel matrix and mixed well at 4°C before coating the Transwell inserts. HTR8/SVneo cells (4.0 × $10^4$ cells) in serum-free RPMI 1640 medium were added to the upper chamber, and RPMI 1640 medium containing 2% FBS was added to the lower well to induce invasion. After incubation at 37°C for 24 h under normoxic conditions (20% $O_2$) or hypoxic conditions (2% $O_2$), noninvading cells were removed using a cotton swab. Transwells were washed and fixed in 4% PFA in PBS at RT for 20 min. Cells on the Transwells were then stained with 0.01% crystal violet. At least 10 regions of interest (ROIs) were randomly set in each Transwell, and the number of invading cells in each ROI was counted via a BA410E-1080M microscope (×3,200 magnification; Shimadzu).

### RNA transcriptome analysis

HTR8/SVneo cells were cultured in serum-free RPMI 1640 medium in the presence or absence of Aβ40 monomers (50 nM) for 24 h. Total RNAs were extracted using PureLink RNA Mini Kit (Thermo Fisher Scientific), and sequencing libraries were generated using NEBNext Ultra RNA Library Prep Kit for Illumina (New England Biolabs). Libraries obtained were purified using the AMPure XP system (Beckman Coulter), and the library quality was assessed on the Agilent Bioanalyzer 2100 system (Agilent). The purified libraries were sequenced on a NovaSeq 6000 (Illumina) with a 150-base pair-end read setting. Sequence qualities were then checked using FastQC (http://www.bioinformatics.babraham.ac.uk/projects/fastqc) and MultiQC (Ewels et al, 2016). Adaptor sequences were removed using Trimmomatic (Bolger et al, 2014), and the poly A/T tail, short sequence length reads, and trim low-quality bases were removed using PRINSEQ-lite (Schmieder & Edwards, 2011). Reads were then aligned to the *Homo sapiens* reference genome (hg38) using STAR (Dobin et al, 2013). Gene expression levels were calculated using the featureCount function from the Subread (Liao et al, 2013), and DEGs were determined using the R package edgeR (Robinson et al, 2010). Gene expressions were normalized to counts per million. Genes expressed in less than two samples were further

removed after the trimmed mean of *M*-value normalization. Principal component analysis was performed from the Euclidean distance using the R/prcomp function, and plotted on a two-dimensional plane. Reactome pathway enrichment analysis was performed using the R package ReactomePA (Yu & He, 2016) and visualized using ClustVis (Metsalu & Vilo, 2015). Genome-wide annotation for Human. R package version 3.8.2 (org.Hs.eg.db) was used as the database for the enrichment analysis.

## Statistical analysis

For statistical analysis, we used an ordinary one-way analysis of variance with the post hoc Dunnett, Bonferroni, or Tukey test, or unpaired *t* test with Prism software (version 7.04; GraphPad Software). The Mann–Whitney test was used for statistical analysis of the quantification in immunohistochemistry. Results were said to be significant when *P*-values were less than 0.05.

# Data Availability

All data are included in the article and/or supporting information. RNA-seq data generated have been deposited in the Sequence Read Archive (BioProject ID: PRJNA1157990).

# Supplementary Information

# Acknowledgements

Thanks are due to Dr. Hiroshi Mori at Nagaoka Sutoku University for his continuous encouragement. This work was partly supported by Grants-in-Aid from the Ministry of Education, Culture, Sports, Science and Technology (MEXT)/Japan Society for the Promotion of Science (JSPS) (23K08853 to K Nishioka, 19K09784 to M Ikezaki, 22K09601 to N Iwahashi, and 20K09605 and 20KK0371 to K Nishitsuji). This study was also partly supported by an internal grant from Wakayama Medical University (Tokutei-kenkyu-zyosei 2019 and 2020 to K Nishitsuji).

## Author Contributions

K Nishioka: formal analysis, funding acquisition, investigation, visualization, and writing—original draft.
M Ikezaki: data curation, formal analysis, funding acquisition, investigation, visualization, and methodology.
N Iwahashi: conceptualization, funding acquisition, investigation, and methodology.
M Arakawa: investigation.
M Fukushima: investigation.
N Mori: investigation.
M Mizoguchi: investigation.
Y Horiuchi-Tanizaki: investigation.
M Fujino: investigation.
T Tomiyama: resources.
Y Ihara: resources.
K Uchimura: conceptualization, resources, supervision, methodology, and writing—review and editing.
K Ino: resources.
K Nishitsuji: conceptualization, resources, data curation, formal analysis, funding acquisition, validation, investigation, visualization, methodology, project administration, and writing—original draft, review, and editing.

## Conflict of Interest Statement

The authors declare that they have no conflict of interest.

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
