## [Reviewer comments · Life Science Alliance]

Amyloid- β fibrils accumulated in preeclamptic placentas suppress cytotrophoblast syncytialization

Kaho Nishioka, Midori Ikezaki, Naoyuki Iwahashi, Miyu Arakawa, Momo Fukushima, Noa Mori, Mika Mizoguchi, Yuko Horiuchi-Tanizaki, Megumi Fujino, Takami Tomiyama, Yoshito Ihara, Kenji Uchimura, Kazuhiko Ino, and Kazuchika Nishitsuji

DOI: <https://doi.org/10.26508/lsa.202503453>

Corresponding author(s): Kazuchika Nishitsuji, Wakayama Medical University

Review Timeline:

Submission Date:	2025-07-14
Editorial Decision:	2025-07-22
Revision Received:	2025-10-21
Editorial Decision:	2025-12-24
Revision Received:	2025-12-25
Accepted:	2026-01-02

Scientific Editor: Sarita Hebbbar

Transaction Report:

Please note that the manuscript was reviewed at *Review Commons* and these reports were taken into account in the decision-making process at *Life Science Alliance*.

Review
COMMONS

Reviews

Review #1

Proving that more Beta-amyloid are produced in preeclampsia, and that impacts negatively trophoblast cell fusion is interesting and provides a potential mechanism for interpreting some specific cases of preeclampsia.

The authors analyzed the placenta from five control and five preeclamptic pregnancies (4 early onset et 1 late onset).

The authors show first by IHC that amyloid beta and aggregate markers are apparently exclusively detected in the PE samples, the same observation is done for detection of HIF1alpha and BACE1, the enzyme that is responsible for the generation of amyloid peptides from digestion of the APP membrane neuron protein.

After having used placental samples, the authors moved to the BeWo cell model, where they could analyze specifically cell biology in the context of syncytialization.

The authors inhibited HIF1a prolylation (thus stabilizing it even in normoxia), and this led to the increase of BACE1, of beta-amyloid molecules, as shown by WB analyses; the same result was obtained by exposure to hypoxia, while a BACE1 inhibitor had the opposite effect.

An interesting issue is the demonstration provided by the authors that in this model, syncytialization is decreased by Beta-amyloid fibrils, together with decreased hCG expression and decreased Syncytin-1. The authors also validate these results on primary human CTB from the third trimester.

****Minor remarks****

1. It is classical now to present in extenso the WB as supplementary data for Fig 3, 4 and 5.
2. It seems that the beta amyloid signal is not stronger for the early onset and the late onset PE samples. Have the authors an interpretation?
3. The figure 4b does not show the BeWo labeling in forskolin with or without beta amyloid peptides, why? It would be illustrative to show a decrease in the fusion processes
4. How do the authors explain that exposure to fibrils did not seem to slow down significantly the fusion process, even though markers are decreased?
5. Could the authors attempt a labeling with the Di-8, an interesting quantitative marker of cell fusion (see ref PMID: 38019394).

This study aims to bridge a gap between the mechanisms of preeclampsia and neurodegenerative disorders, and this through the existence of misfolded proteins in the preeclamptic placenta which has been reported before, in particular the beta amyloid protein, known as operative in Alzheimer's disease (AD) in particular.

Review #2

In this manuscript, the authors examine the deposition of amyloid- β (A β) peptides that accumulate in the brains of patients with Alzheimer's disease (AD). The authors demonstrated the expression of HIF-1 α in the pre-eclamptic (PE) placental tissue using immunofluorescence (which is not novel), alongside the expression of BACE1. These experiments were also validated using BeWo and primary trophoblast cells cultured under hypoxia to mimic one of the characteristics of PE. However, this manuscript is quite preliminary, and many additional experiments are necessary to confirm the deposition of A β fibrils in PE. The authors treated CTB and observed the effects on STB, but in PE, the main cell lineage affected is extravillous trophoblast (EVT) cells, which invade the spiral artery. The defect in this invasion is one of the major causes of PE. Therefore, the authors should investigate the effect of hypoxia and A β deposition on EVT invasion. Overall, this work appears very incomplete, and further experiments are warranted.

****Major comments****

- If CTBs are treated with A β , and if it affects STB, what happens with EVT? Why didn't they check with EVT if the authors wanted to link with PE?
- Did the authors look for pathologies related to A β deposition on PE placentas?
- Line# 103, the IF images don't show that BACE1 is around HIF1 α . There are no merged images, and the results are over- or underestimated.

- What is the intended purpose of using Roxadustat? If it inhibits HIF1 α , could you explain the reason behind the increased expression of HIF1 α ? Furthermore, is there evidence to support the efficacy of this compound?
- Is A β deposition very specific to PE, or can it also occur for other reasons during pregnancy?
- BACE1 is expressed in Normal#2 and #3 but not in #1, #4, and #5. Why is this expressed in #2 and #3? Is there anything wrong with these samples? If patients had gestational hypertension or some other complications?
- PE placentae were compared with GA matched placentae. What is the expression of BACE1 and RB4CD12 in term control placentae?
- If AB fibril deposition is hypoxia dependent, what happens at the early gestation, where oxygen conc is 1-2%?

****Minor comments****

- The authors only performed IF and IHC. Please confirm and correct the methods accordingly.
- Was the BeWo-b21 clone cell line used for all the experiments in this paper? This is the only clone that can be used for BeWo-STB models.
- Have all the experiments on BeWo only been performed once?

Investigating the deposition of A β in the placenta could enhance our understanding of pregnancy complications such as PE, fetal growth restriction, and neurodevelopmental risks. However, further research on this topic is necessary.

Review #3

The authors found that Amyloid β suppressed cytotrophoblasts syncytialization, which is innovative. The authors used human patient samples and human primary CTB culture which are powerful data.

Fig. 3. The authors used Roxadustat to stimulate HIF-1 α and showed BACE1 increase. It would be better to have the cells in real hypoxia condition.

Fig. 4 and 5. The authors used external Amyloid β for stimulation. Would the endogenous Amyloid β levels reach the concentration of external one? It would be better to see the quantitative levels of Amyloid β in Fig. 3b.

While A β is present in human placentas and accumulates in preeclamptic placentas, the production and role of A β in the human placenta remain unclear. The current findings suggest that increased A β production in cytotrophoblast by hypoxia may lead to the formation of A β fibrils, which inhibit syncytiotrophoblast formation and are detrimental to pregnancy, revealing a novel role of A β fibrils in the pathogenesis of preeclampsia.

Review #4

This manuscript focuses on the role of amyloid β (A β) in hypoxia-exposed human trophoblasts. Recent reports in the literature have confirmed the presence A β and other proteins, including Tau, transthyretin, and TDP-43, in placental tissue derived from preeclampsia deliveries. These proteins are recognized as hallmark causative factors for Alzheimer's disease related dementias. Hypoxia has also been shown to induce expression of these proteins, including A β , in human trophoblasts. In this regard, detection of A β hypoxia-exposed human trophoblast may not be a novel finding. This said, the manuscript presents some solid information and could have been very informative study. However, several conceptual, technical and literature concerns remain unaddressed and dampen the reviewer's enthusiasm for this study.

****Major comments:****

1. In lines 50 and 51 of Introduction, the authors provide references to two publications. However, several other articles have appeared before or after these publications that demonstrated evidence for proteinopathy in the placenta and circulation of preeclampsia patients. The reviewer has gone through the literature and found several publications. For example, Kalkunte et al were the first ones to demonstrate the etiology of proteinopathy in preeclampsia placenta and focused on a protein called transthyretin (Am J Pathol. 2013, 183(5):1425-1436). Similarly, Cheng et al demonstrated using a novel blood test that serum from early onset and late onset preeclampsia manifestations contained A β and transthyretin (Nature Sci Rep. 2021;11:15934). Jash et al showed the presence of cis P-tau in the placenta and serum of early and late onset preeclampsia patients (Nat Commun. 2023;14:5414). This article and another article by Cheng et al (Hypertension 79(8):1738-1754) revealed that aggregated cis P-tau and transthyretin

are etiologically critical for the onset of preeclampsia. There have been several other review and original articles that have talked about Alzheimer's like etiology in preeclampsia (Olie et al, JAMA Netw Open, 2024; e2412870; Basit et al, BMJ 2018; 363:k4109; Schliep et al, Hypertension 2023; 80:257-267, Cheng et al, Am J Reprod Immunol. 2016;75:372-381).

2. Following up on the comments made above, the authors talk about induction of A β in hypoxia-treated human trophoblasts represented by an established cell line, BeWo, and primary human trophoblasts. However, it is not clear whether A β 42 as stated in the manuscript was detected as an aggregated structure or a protein coupled with RB4CD12 aggregate marker. It would have been helpful if the authors could provide direct evidence for A β aggregation.

3. What appeared to be more surprising is the statement on lines 162 and 163 that cultured CTBs produced A β 40/42. Again, it is not clear whether the authors are talking about aggregated A β or just induction of A β . Why should normal CTBs produce A β ? It is not clear whether this is a transient expression or a long-term phenomenon. The issue is distinction between normal and adverse pregnancy conditions, and the latter associated with protein aggregation as suggested in the literature.

4. The authors have adequately pointed to importance of hypoxia in the onset of preeclampsia-like features. As a matter of fact, Lai et al demonstrated in a mouse pre-clinical model that hypoxia could induce severe features of preeclampsia (Hypertension. 2011;57:505-514). The use of hypoxia as driver of A β induction is appreciated.

5. In Fig. 1, although the authors have used DIC approach, it would have been helpful if they presented individual A β and RB4CD12 green and red channels, and a merged profile. For example, PE #4 sample does not appear to have much RB4CD12. Again, there is a question of aggregated or native protein structures. It is difficult to have a satisfactory statistical analysis. Did the authors look for A β in the anchoring villi region of the placenta?

6. Fig. 2 does not show significant staining for HIF1- α in PE placental tissue.

7. Fig. 3B, why should there be A β 40/42 under normoxic conditions? This is the most pertinent concern and the authors are validating significant expression of A β 40/42 under normal conditions. In normal pregnancy placenta, this protein has not been detected.

8. Figs. 4 and 5 present the crux of the conclusions that the authors are trying to draw from their study. A β peptide solution was incubated for 5 days at 37°C to prepare so called A β fibril-like structures. What is the purity of fibril structures? Does this preparation show toxic effects on cell viability? Human trophoblasts expressing E-cadherin fail to participate in endovascular cross-talk with endothelial cells, a process required for spiral arteries. It appears that either BeWo cells or primary trophoblasts used in this study represent trophoblasts from third trimester. It is not clear why should A β fibril like structures should inhibit ZO-1 and E-cadherin or β -hCG (Fig. 5) for that matter. In Fig. 5C, there does not seem to be a major effect of A β fibrils. Did the authors try synthetic A β as a control. These experiments could have been meaningful but for proper controls.

The manuscript addresses an important theme recently identified to address the heterogeneous etiology of preeclampsia. Although the authors have used in vitro approaches, the study could have been a solid if not for some major concerns.

The authors have focused on an already demonstrated phenomenon but have tried to validate the findings using their in vitro approaches. The manuscript is well written but some lapses for correct references.

July 22, 2025

Re: Life Science Alliance manuscript #LSA-2025-03453

Kazuchika Nishitsuji
Wakayama university
Unknown
Japan [JP]

Dear Dr. Nishitsuji,

Thank you for submitting your manuscript entitled "Amyloid- β fibrils accumulated in preeclamptic placentas suppress syncytialization of cytotrophoblasts", and the associated referee evaluation reports from Review Commons, to Life Science Alliance.

We invite to submit a manuscript revised according to your revision plan.

Thank you for this interesting contribution to Life Science Alliance. We are looking forward to receiving your revised manuscript.

Sincerely,

Sarita Hebbar, PhD
Scientific Editor
Life Science Alliance
<http://www.lsjournal.org>

B. MANUSCRIPT ORGANIZATION AND FORMATTING:

Point-by-point response to the Reviewers' comments

Reviewer #1 (Significance (Required)):

This study aims to bridge a gap between the mechanisms of preeclampsia and neurodegenerative disorders, and this through the existence of misfolded proteins in the preeclamptic placenta which has been reported before, in particular the beta amyloid protein, known as operative in Alzheimer's disease (AD) in particular.

Our response: We sincerely appreciated the reviewer's constructive comments.

Reviewer #1 (Evidence, reproducibility and clarity (Required)):

Minor remarks

1. *It is classical now to present in extenso the WB as supplementary data for Fig 3, 4 and 5.*

Our response: We included the full blots as Source data files.

2. *It seems that the beta amyloid signal is not stronger for the early onset and the late onset PE samples. Have the authors an interpretation?*

Our response: The current manuscript includes both early-onset and late-onset cases. Thus, we are certain that amyloid beta deposition is involved in both early- and late-onset PE pathologies. Our immunohistochemical analysis with an anti-HIF1- α revealed that PE placentas are under chronic hypoxia, which may lead to a sustained increase in the production and local concentration of A β peptides and fibrillization of A β . **We now describe these points on page 7, lines 126–132.**

Page 7, lines 126–132

The current study includes four early-onset and one late-onset PE cases (Table 1). The placentas of patients with early- and late-onset PE are under hypoxic and hypo-perfusion conditions (Baylis, Zhou, Menkhorst, & Dimitriadis, 2024; Soto et al., 2012). Our immunohistochemical analysis revealed that early- or late-onset PE placentas were in a hypoxic condition, which may lead to a sustained increase in the production and local concentration of A β peptides and fibrillization of A β . Thus, these results support that A β deposition may be involved in the both early- and late-onset PE pathologies.

3. *The figure 4b does not show the BeWo labeling in forskolin with or without beta amyloid peptides, why? It would be illustrative to show a decrease in the fusion processes*

Our response: In Fig. 4A, we pretreated BeWo cells with A β fibrils and after that, cell fusion was induced by Fsk. On the other hand, in Fig. 4B, we treated BeWo with A β fibrils and investigated the protein levels and subcellular localization of ZO-1 and E-cadherin. Fig.4b shows that expressions of proteins involved in cell-cell interaction were reduced by A β fibril treatment without Fsk. Cell-cell interaction before syncytialization is required for cell fusion, and these proteins disappear after cell fusion. Thus, our results demonstrate that elimination of cell-cell interaction by A β fibrils resulted in reduced cell fusion induced by Fsk. This is why we treated BeWo cells with A β fibrils before the induction of cell fusion by Fsk, and BeWo labeling in forskolin with or without A β fibrils will result in a loss of ZO-1 and E-cadherin regardless of the occurrence of cell fusion. **We described these points in more detail on page 10, lines 182–187, and on page 10, lines 198–200.**

Page 10, lines 182–187

A β aggregates disrupt membrane localization of tight junction proteins, at least in part, by inducing excess autophagy (Chan et al., 2018; Marco & Skaper, 2006). We hypothesized that A β fibrils might also disrupt the membrane localization of ZO-1 and E-cadherin in cytotrophoblasts. Therefore, we next investigated the effect of A β 42 fibrils on the subcellular localization of ZO-1 and E-cadherin in BeWo cells and how this affects CTB syncytialization.

Page 10, lines 198–200

these results suggest that A β 42 fibrils interfered with Fsk-induced syncytialization by disrupting the proper membrane localization of cell adhesion-related proteins that are required for syncytialization.

4. *How do the authors explain that exposure to fibrils did not seem to slow down significantly the fusion process, even though markers are decreased?*

Our response: Since we previously demonstrated that loss of membrane E-cadherin slows the fusion (Iwahashi et al., Endocrinology, 2019, PMID: 30551188), we believe that reduction of membrane localization of E-cadherin also slows the fusion process. **We describe this point on page 15, lines 284–286.**

Page 15, lines 284–286

Since the loss of membranous E-cadherin was found to slow the fusion process in BeWo cells (Iwahashi et al., 2019), it is possible that A β fibrils also slow the CTB fusion process.

5. Could the authors attempt a labeling with the Di-8, an interesting quantitative marker of cell fusion (see ref PMID: 38019394).

Our response: We have shown that pretreatment of BeWo cells and human primary cytotrophoblasts (CTBs) inhibited induction of syncytin-1 and β -hCG. Syncytin-1 is a critical driver of syncytialization and formation of the syncytiotrophoblast layer, and β -hCG is one of the major products of syncytiotrophoblasts. Thus, induction of these proteins is widely used as syncytialization markers of trophoblasts. On the other hand, Di-8-ANEPPS is a potentiometric fluorescent dye that may be used to assess cell fusion simply and economically. Although we understand the robustness of this method, we believe that the current data are sufficient to demonstrate that A β fibril pretreatment inhibited syncytialization of BeWo cells and CTBs.

Reviewer #2 (Significance (Required)):

Investigating the deposition of A β in the placenta could enhance our understanding of pregnancy complications such as PE, fetal growth restriction, and neurodevelopmental risks. However, further research on this topic is necessary.

Our response: We sincerely appreciate the critical reading and constructive comments of the reviewer. We agree that further research on protein aggregation and the pathogenesis of preeclampsia is necessary. **We discuss this matter in the discussion on page 13, line 260–page 14, line 263, and on page 15, lines 291–294.**

Page 13, line 260–page 14, line 263

Currently, the production of A β across gestation has not been studied, and the stage at which A β begins to deposit in PE placentas is unclear. Additional studies are necessary to elucidate the significance of A β metabolism and deposition in the physiology and pathology of the human placenta.

Page 15, lines 291–294

It is notable that endoplasmic reticulum stress and oxidative stress were reportedly induced in PE placentas (Raijmakers et al., 2004; Veerbeek, Tissot Van Patot, Burton, & Yung, 2015). Further investigation is necessary to identify the stress involved in A β deposition in PE placentas.

Reviewer #2 (Evidence, reproducibility and clarity (Required)):

Major comments

1. If CTBs are treated with A β , and if it affects STB, what happens with EVT? Why didn't they check with EVT if the authors wanted to link with PE?

Our response: We thank the reviewer for the critical comment. We investigated A β generation by an EVT model cell, HTR8/SVneo cells. We found that HTR8/SVneo cells produced much less amount of A β compared to BeWo cells (**please see on page 8, line 159–page 9, line 164**). Gao et al. reported that A β aggregates induced autophagy in HTR8/SVneo cells and suggested that an excessive autophagy may be detrimental and be involved in the development of preeclampsia (Gao et al., J Mol Histol, 2024, PMID: 38777993). **We discussed this matter on page 13, lines 243–245.**

On the other hand, we investigated the effects of A β monomers on EVT, and discovered that even low levels of A β produced by EVT promoted EVT invasiveness. We also performed RNAseq of A β monomer-treated HTR/SVneo cells for clarifying the genes and pathways that were involved in the enhanced invasiveness by A β . **We added these new data as newly prepared Supplemental Figures S5 and S9, and discussed on page 13, lines 243–258.**

Page 8, line 159–page 9, line 164

Although our ELISA failed to detect A β 40 in the conditioned medium (CM) of the EVT model HTR8/SVneo cells (Graham et al., 1993), we found that HTR8/SVneo cells produced A β peptides and that the A β production was enhanced by hypoxia in the immunoblot with the β 001 antibody (Supplemental Figure S5). These results suggested that HTR8/SVneo cells produced much less amount of A β than BeWo cells.

Page 13, lines 243–258 (including page 13, lines 243–245)

A β aggregates reportedly enhanced autophagy in HTR8/SVneo cells and an excessive autophagy may exacerbate PE (L. Gao et al., 2015; Q. Gao et al., 2024). On the other hand, immediately after the implantation, an embryo has to survive a severe low oxygen tension condition due to the lack of vasculature and oxygen supply (Pollheimer, Vondra, Baltayeva, Beristain, & Knofler, 2018; Rodesch, Simon, Donner, & Jauniaux, 1992). Thus, EVT aggressively invade the uterine decidua and myometrium to remodel the maternal spiral arteries and develop a vasculature for the maternal-fetal interface (Pijnenborg, Dixon, Robertson, & Brosens, 1980; Tayade, Black, Fang, & Croy, 2006). Here, we also found that HTR8/SVneo cells, a widely accepted model of EVT (Graham et al., 1993), produced much less amount of A β than BeWo cells and nM range of A β promoted EVT invasiveness (Supplemental Figure S9). We identified 1444 differentially expressed genes in A β -treated HTR8/SVneo cells and our transcriptome analysis suggested that quite low concentration of A β enhanced EVT invasiveness by activating R-HSA-1442490 (collagen degradation) and R-HSA-1592389 (activation of matrix metalloproteases). These results suggest that while A β fibrils are detrimental to the placenta, nM range of A β monomers may have a physiological function in early pregnancy.

2. *Did the authors look for pathologies related to A β deposition on PE placentas?*

Our response: We did not observe any notable pathologies within the vicinity of the A β deposition. **We discussed this point in the discussion on page 17, lines 331–332.**

Page 17, lines 331–332

It further remains to be analyzed whether A β deposition in the placenta directly contributes to the pathogenesis of PE.

3. *Line# 103, the IF images don't show that BACE1 is around HIF1. There are no merged images, and the results are over- or underestimated.*

Our response: We agree with the reviewer. There is a time discrepancy between HIF activation and BACE1 induction. Our immunohistochemical analysis showed that PE placentas are in a chronic hypoxia condition and that BACE1 was increased in PE placentas. Our cell-based assay supports that HIF1 α stabilization by Roxadustat increased BACE1 levels in BeWo cells. **We toned down the results section**

of the immunohistochemical analysis by deleting the sentences “Signal intensities of HIF1- α in normal placentas were weak and significantly lower than those of PE placentas, which was consistent with a previous report (Caniggia & Winter, 2002)”. The results sections are now revised as on page 6, line 117–page 7, 123.

Page 6, line 117–page 7, line 123

Because PE placentas exist under hypoxic conditions (Tong et al., 2022), we analyzed the expression of HIF1- α in human normal and PE placentas. In a normoxic condition, HIF1- α is constitutively expressed but degraded via the proline-hydroxylation and the subsequent ubiquitination and degradation in the proteasome. Because the proline-hydroxylation is oxygen-dependent, hypoxia induce HIF1- α accumulation (Masson & Ratcliffe, 2003). Immunohistochemistry revealed the induction of HIF-1 α in PE placentas, suggesting a hypoxic environment consistent with a previous study (Caniggia & Winter, 2002).

4. What is the intended purpose of using Roxadustat? If it inhibits HIF1 α , could you explain the reason behind the increased expression of HIF1 α ? Furthermore, is there evidence to support the efficacy of this compound?

Our response: Roxadustat inhibits the proline hydroxylation of HIF1 α and thereby inhibits the ubiquitination and degradation of HIF1 α via the ubiquitin proteasomal system (reviewed in Su et al., Drug Discov Today, 2020, PMID: 32380083). In this study, we used Roxadustat as a HIF1 α stabilizer to investigate whether BACE1 levels are increased with hypoxia and HIF1. Our data showed that treatment of BeWo cells with Roxadustat increased HIF1 α levels, supporting the efficacy of Roxadustat. **We describe the use of roxadustat as a HIF1- α stabilizer on page 7, line 141–page 8, line 144.**

Page 7, line 141–page 8, line 144

In order to study the effect of HIF-1 α on BACE1 expression, we used the HIF1- α stabilizer roxadustat (Su, Li, Yu, & Zhang, 2020), which inhibits the prolyl hydroxylation and subsequent degradation of HIF1- α (Hsieh et al., 2007).

5. Is A β deposition very specific to PE, or can it also occur for other reasons during pregnancy?

Our response: To date, no report has been found showing A β deposition in placentas other than PE. The deposition of protein aggregates, including those of A β and transthyretin, has previously been reported in PE. However, the presence and role of these protein deposits in placentas under pathological conditions, in addition to PE, remains to be elucidated. Several stresses such as hypoxia and ER stress may lead to deposition of protein aggregates in the placenta. **We discussed this point on page 15, lines 291–294, and on page 16, lines 303–307.**

page 15, lines 291–294

It is notable that endoplasmic reticulum stress and oxidative stress were reportedly induced in PE placentas (Raijmakers et al., 2004; Veerbeek, Tissot Van Patot, Burton, & Yung, 2015). Further investigation is necessary to identify the stress involved in A β deposition in PE placentas.

Page 16, lines 303–307

To date, no reports have been found showing A β deposition in placentas other than those from PE. Although the deposition of protein aggregates, including those of A β and TTR, has been reported in PE, the presence and role of these protein deposits in the placenta under physiological or pathological conditions remained to be elucidated.

6. BACE1 is expressed in Normal#2 and #3 but not in #1, #4, and #5. Why is this expressed in #2 and #3? Is there anything wrong with these samples? If patients had gestational hypertension or some other complications?

Our response: We did not find any other complications in the normal placentas. In the brain, A β is constitutively generated and thus, thought to play physiological roles. The amount of A β is determined by the balance between the production and the clearance. A sustained imbalance of A β production and A β clearance will lead A β aggregation and deposition. We found that BeWo cells expressed BACE1 in a normoxic condition and thus, normal placentas may express BACE1 and generate small amounts of A β . Our results suggested that chronic hypoxia in PE placentas resulted in increased BACE1 expression and increased A β production, which may eventually result in A β aggregation and deposition, because the aggregation process of A β is concentration-dependent. **We discussed this point in the discussion on page 16, lines 307–317.**

Page 16, lines 307–317

We found that A β is produced by CTBs and EVT s in a basal state. The amount of A β is determined by the balance between the production and the clearance (Selkoe & Hardy, 2016). A sustained imbalance in the production and clearance of A β leads to its aggregation and deposition. Although non-aggregated A β did not affect the syncytialization in BeWo cells, nM or pM A β promoted EVT invasion. This suggests that A β has a physiological function in the placenta. On the other hand, A β fibrils were detrimental to CTBs and EVT s . Thus, normal placentas may express BACE1 and produce small amounts of A β . Our results suggest that chronic hypoxia in PE placentas results in increased BACE1 expression and sustained A β production. Eventually, this leads to A β aggregation and deposition because the aggregation process depends on the local A β concentration. Elucidating the physiological roles of A β in CTB or STB functions deserves future investigation.

7. PE placentae were compared with GA matched placentae. What is the expression of BACE1 and RB4CD12 in term control placentae?

Our response: We used RB4CD12 as a protein aggregation marker. As shown in Table 1, the current study includes 3 placentas whose gestational ages are over 37 weeks. We did not observe RB4CD12 and A β deposition in gestational age-matched control and observed BACE1 expression in one 37 weeks gestational age control. **We also performed additional staining with the ProteoStat protein aggregate detection dye in order to confirm A β fibril deposition in PE placentas (revised Fig. 1).** Again, we did not observe A β deposition in normal placentas. **We described these points in the result section on page 5, line 101–page 6, line 108, and on page 7, lines 133–135.** During the preparation of our revised manuscript, we noticed that we missed include that we used the TrueVIEW Autofluorescent Quenching Kit for quenching the endogenous autofluorescence derived from red blood cells. **We now include the use of these dye and kit in the methods on page 20, lines 389–399.** We would like to thank the reviewers for providing us with the opportunity to correct our mistake. The original Fig.1 shows that A β was co-deposited with heparan sulfate S-domains. Because we think that interactions between A β and biomolecules including heparan sulfate are important for A β aggregation and deposition, **we keep the original Fig.1 as Supplemental Fig. S1.**

Page 5, line 101–page 6, line 108

We used the ProteoStat Protein Aggregation Assay kit for analysis of aggregated A β . A β peptides are small peptides consisting of 40 to 42 amino acid residues that are released extracellularly after production.

Non-deposited A β monomers are not detected by our immunohistochemical analysis, because these soluble A β peptides are spread out in the tissue fluid. Thus, we calculated only merged signals of A β and the ProteoStat dye in order to show aggregated and deposited A β peptides in the placenta. Here, we found that significant deposition of aggregated A β in the villi of five PE cases but not in normal placentas (Figure 1).

Page 7, lines 133–135

Although we detected BACE1 in normal placentas, A β did not deposit in these placentas. As mentioned above, non-deposited soluble A β peptides in normal placentas are thought to spread out in the tissue fluid and were not detected by our immunohistochemical analysis.

Page 20, lines 389–399 (including page 20, lines 392–395 for the TrueVIEW kit)

For detection of A β aggregates, sections were washed with PBS, incubated in ProteoStat solution (1:2,000 in the ProteoStat assay buffer) for 3 min, and then destained in 1% acetic acid for 20 min at room temperature (Matafora et al., 2020). Endogenous autofluorescence derived from red blood cells were quenched by using the TrueVIEW Autofluorescence Quenching Kit (Vector Laboratories) and the sections were mounted with Vectashield mounting medium with 4',6-diamidino-2-phenylindole (DAPI) (Vector Laboratories). Specimens were then examined with an LSM700 confocal microscope (Carl Zeiss, Jena, Germany). For quantification, 5 regions of interest (ROIs; 16 $\mu\text{m} \times 16 \mu\text{m}$) were set on villi in each placenta tissue, and mean intensities were determined by using ZEN 3 blue edition (Carl Zeiss). β 001- and ProteoStat-positive areas were determined by using the Coloc. Tools of ZEN 3.

8. If AB fibril deposition is hypoxia dependent, what happens at the early gestation, where oxygen conc is 1-2%?

Our response: At the early gestation, physiological hypoxia promotes the EVT invasion and helps the remodeling of spiral arteries for oxygen supply. **Please see our response to Comment 1.** Severe hypoxia on the CTB side in early gestation may result in a miscarriage before PE develops.

Minor comments

1. The authors only performed IF and IHC. Please confirm and correct the methods accordingly.

Our response: We thank the reviewer for pointing this out. In the present study, we performed immunohistochemical analysis of placental tissues, immunofluorescent analysis and immunoblot analysis of BeWo cells and human primary CTBs. We included methods of these experiments in the methods. We also included the methods for A β fibril preparation and human primary CTB preparation. **We deleted the unnecessary description “anti-human A β 1-42 rabbit IgG affinity-purified antibody” and “Anti-amyloid β 40 mouse monoclonal antibody BA27 was obtained from Wako Pure Chemicals (Osaka, Japan)” in the materials.**

2. Was the BeWo-b21 clone cell line used for all the experiments in this paper? This is the only clone that can be used for BeWo-STB models.

Our response: We do not have information about the clone number of BeWo cells used in this study. We purchased them from the American Type Culture Collection (Manassas, VA) and they were authenticated by JCRB Cell Bank (National Institute of Biomedical Innovation Japan, report no. KBN0410). By using the same cells, we published three articles in which we successfully analyzed syncytialization of BeWo cells (Yamamoto et al., Endocrinology, 2017, PMID: 28938427; Iwahashi et al., Endocrinology, 2019, PMID: 30551188; Matsukawa et al., Biomolecules, 2022, PMID: 36008943). We would like to apologize

for our mistake in the description of BeWo cells in the methods section and thank the reviewer for providing us with an opportunity to correct our mistake. **We noted that BeWo cells were purchased from the American Type Culture Collection (Manassas, VA) and authenticated by JCRB Cell Bank (National Institute of Biomedical Innovation Japan, report no. KBN0410) in the method section on page 21, lines 401–405, and uploaded the authentication report KBN0410 as a review process file.**

Page 21, lines 401–405

Human choriocarcinoma-derived BeWo cells were purchased from the American Type Culture Collection (Manassas, VA) and they were authenticated by JCRB Cell Bank (National Institute of Biomedical Innovation Japan, report no. KBN0410). We successfully analyzed syncytialization by using them (Iwahashi et al., 2019; Iwahashi et al., 2021; Yamamoto et al., 2017).

3. Have all the experiments on BeWo only been performed once?

Our response: We repeated 6 experiments (the repetitions are biological, not technical, replicates). The results are shown as means \pm SEM (n = 6) as stated in the Figure legends.

Reviewer #3 (Significance (Required)):

While A β is present in human placentas and accumulates in preeclamptic placentas, the production and role of A β in the human placenta remain unclear. The current findings suggest that increased A β production in cytotrophoblast by hypoxia may lead to the formation of A β fibrils, which inhibit syncytiotrophoblast formation and are detrimental to pregnancy, revealing a novel role of A β fibrils in the pathogenesis of preeclampsia.

Reviewer #3 (Evidence, reproducibility and clarity (Required)):

The authors found that Amyloid β suppressed cytotrophoblasts syncytialization, which is innovative. The authors used human patient samples and human primary CTB culture which are powerful data.

Our response: We appreciate the reviewer's thoughtful feedback and support.

1. Fig. 3. The authors used Roxadustat to stimulate HIF-1 α and showed BACE1 increase. It would be better to have the cells in real hypoxia condition.

Our response: Because the purpose of this experiment is to show that increased HIF1- α correlated BACE1 induction, we used Roxadustat as a HIF1- α stabilizer and showed that sustained induction of HIF increased BACE1 levels. However, we do understand the reviewer's concern. **We performed Western blotting showing increases in HIF1- α and BACE1 in hypoxic conditions. These new results are now included as Fig. 3A and we mentioned on this point on page 7, lines 139–140. We also revised the methods on page 21, lines 417–420.**

Page 7, lines 139–140

Hypoxia treatment increased HIF1- α and BACE1 protein levels (3.8-fold for HIF1- α and 1.5-fold for BACE1, Fig. 3A).

Page 21, lines 417–420

The cells were then cultured for an additional 3 or 24 h in hypoxic conditions. HIF1- α and BACE1 protein levels were analyzed by Western blotting using an anti-HIF-1 α antibody (1:1000) and an anti-human BACE1 (C) antibody (1:50), respectively, as described below.

[Figure removed by editorial staff per authors' request].

2. Fig. 4 and 5. The authors used external Amyloid β for stimulation. Would the endogenous Amyloid β levels reach the concentration of external one? It would be better to see the quantitative levels of Amyloid β in Fig. 3b.

Our response: We performed ELISA assays to quantitatively analyze A β generation by BeWo cells. We included the ELISA data as newly prepared Supplemental Figure S3 and mentioned in the results on page 8, lines 150–153 and on page 8, line 159–page 9, line 164. Because the aggregation of A β requires a high concentration of a micromolar order, we used synthetic A β fibrils for stimulation. We propose that chronic hypoxia in preeclampsia leads to an elevated local concentration of A β through a sustained increase in A β production, which eventually results in A β fibrillogenesis and deposition of A β fibrils. Therefore, it will be difficult for the A β concentrations generated by BeWo cells to reach a level sufficient for fibrillogenesis *in vitro*. We also found that A β was co-deposited with highly sulfated domains of heparan sulfate, which reportedly co-deposited with several protein aggregates such as those of A β , transthyretin, and tau, and promote aggregation of those proteins (Hosono et al., Am J Pathol, 2012, PMID: 22429964; Kameyama et al., Am J Pathol, 2019, PMID: 30414409; Castillo et al., J Neurochem, 1999, PMID: 10098877; Nishitsuji and Uchimura, Glycoconj J, 2017, PMID: 28401373). Sustained increase in the local concentration of A β and interaction between A β with these biomolecules

are important for fibrillogenesis of A β and subsequent deposition. We also think that investigating the effects of ER stress on A β aggregation is another important future research topic. **We discussed this point in the discussion on page 15, lines 291–294 and on page 16, line 317–page 17, line 329.**

Page 8, lines 150–153

According to our enzyme-linked immunosorbent assay (ELISA), A β 40 production increased 126% under hypoxic conditions compared with A β 40 production under normoxic conditions, and it decreased, 44% and 34%, after use of the LY2886721 BACE1 inhibitor under both normoxic and hypoxic conditions, respectively (Supplemental Figure S3).

Page 8, line 159–page 9, line 164

Although our ELISA failed to detect A β 40 in the conditioned medium (CM) of the EVT model HTR8/SVneo cells (Graham et al., 1993), we found that HTR8/SVneo cells produced A β peptides and that the A β production was enhanced by hypoxia in the immunoblot with the β 001 antibody (Supplemental Figure S5). These results suggested that HTR8/SVneo cells produced much less amount of A β than BeWo cells.

Page 15, lines 291–294

It is notable that endoplasmic reticulum stress and oxidative stress were reportedly induced in PE placentas (Raijmakers et al., 2004; Veerbeek, Tissot Van Patot, Burton, & Yung, 2015). Further investigation is necessary to identify the stress involved in A β deposition in PE placentas.

Page 16, line 317–page 17, line 329

The aggregation processes of proteins, including those of A β 42, are concentration-dependent and require concentrations higher than the micromolar range (Burdick et al., 1992; Knowles, Vendruscolo, & Dobson, 2014). Because amyloid fibrils and A β are in equilibrium, A β concentrations around amyloid deposits are expected to be quite high (DeMattos et al., 2002). Therefore, we propose that chronic hypoxia in PE leads to an elevated local concentration of A β through a sustained increase in its production. This ultimately results in A β fibrillogenesis and the deposition of A β fibrils. In addition, several biomolecules such as sulfated glycosaminoglycans are known to be critical for fibrillogenesis of amyloidogenic proteins (K. Nishitsuji & Uchimura, 2017). Accordingly, A β co-deposited with highly sulfated heparan sulfate (Supplemental Figure S1), which supports the involvement of sulfated glycosaminoglycans in placenta A β deposition. Further investigation of this topic is another future challenge.

Reviewer #4 (Significance (Required)):

The manuscript addresses an important theme recently identified to address the heterogeneous etiology of preeclampsia. Although the authors have used in vitro approaches, the study could have been a solid if not for some major concerns.

The authors have focused on an already demonstrated phenomenon but have tried to validate the findings using their in vitro approaches. The manuscript is well written but some lapses for correct references.

Our response: We thank the reviewer for the critical reading of our manuscript and his/her constructive comments. As the reviewer pointed out, recent studies suggest that preeclampsia is a proteinopathy. However, the mechanisms by which protein aggregate plays detrimental roles in placentation has not been well-understood. In the present study, we discovered a detrimental role of A β fibrils in syncytiotrophoblast formation.

Reviewer #4 (Evidence, reproducibility and clarity (Required)):

Major comments:

1. In lines 50 and 51 of Introduction, the authors provide references to two publications. However, several other articles have appeared before or after these publications that demonstrated evidence for proteinopathy in the placenta and circulation of preeclampsia patients. The reviewer has gone through the literature and found several publications. For example, Kalkunte et al were the first ones to demonstrate the etiology of proteinopathy in preeclampsia placenta and focused on a protein called transthyretin (Am J Pathol. 2013, 183(5):1425-1436). Similarly, Cheng et al demonstrated using a novel blood test that serum from early onset and late onset preeclampsia manifestations contained A β and transthyretin (Nature Sci Rep. 2021;11:15934). Jash et al showed the presence of cis P-tau in the placenta and serum of early and late onset preeclampsia patients (Nat Commun. 2023;14:5414). This article and another article by Cheng et al (Hypertension 79(8):1738-1754) revealed that aggregated cis P-tau and transthyretin are etiologically critical for the onset of preeclampsia. There have been several other review and original articles that have talked about Alzheimer's like etiology in preeclampsia (Olie et al, JAMA Netw Open, 2024; e2412870; Basit et al, BMJ 2018; 363:k4109; Schliep et al, Hypertension 2023; 80:257-267, Cheng et al, Am J Reprod Immunol. 2016;75:372-381).

Our response: We sincerely appreciate the reviewer for his/her helpful comment. **We revised the introduction by citing the references recommended by the reviewer on page 3, line 51–page 4, line 68.**

Page 3, line 51–page 4, line 68

The abnormal deposition of misfolded proteins, such as transthyretin and Thr231-phosphorylated *cis*-P-tau, has been implicated in the etiology and pathology of PE (S. Cheng et al., 2021; S. Cheng et al., 2022; Jash et al., 2023; Kalkunte et al., 2013). In addition, several studies reported deposition of aggregated amyloid β (A β) peptides in PE placentas (Buhimschi et al., 2014; Cater et al., 2019; S. Cheng et al., 2021). A β peptides, which deposit in the brains of patients with Alzheimer's disease (AD), is produced by the sequential cleavage of amyloid precursor protein (APP) by β -secretase 1 (BACE1) and γ -secretase. A β levels in the brain are determined by the balance between A β production and clearance (Selkoe & Hardy, 2016). Thus, an imbalance between A β production and clearance leads to increased A β levels as well as A β aggregation to form toxic A β aggregates. APP is widely expressed throughout the body, including the placenta, and can be processed by BACE1 and γ -secretase to produce A β in the placenta (Buhimschi et al., 2014; Mattson, 2004). Association between hypertensive disorders in pregnancy including PE and dementia and the involvement of aggregation of proteins in PE have been documented (Basit, Wohlfahrt, & Boyd, 2018; S. B. Cheng, Nakashima, & Sharma, 2016; Olie et al., 2024; Schliep et al., 2023), which supports that PE is a placental proteinopathy. A β aggregates have been established as toxic to neurons (Selkoe & Hardy, 2016), however, exactly how A β and A β aggregates affect placental cell functions is unknown.

2. Following up on the comments made above, the authors talk about induction of A β in hypoxia-treated human trophoblasts represented by an established cell line, BeWo, and primary human trophoblasts. However, it is not clear whether A β 42 as stated in the manuscript was detected as an aggregated structure or a protein coupled with RB4CD12 aggregate marker. It would have been helpful if the authors could provide direct evidence for A β aggregation.

Our response: Based on our previous findings showing that highly sulfated domains of heparan sulfate are common components of protein aggregate deposits, we used RB4CD12, which recognizes these domains, as a marker of protein aggregate deposition. These include aggregates of A β in Alzheimer's disease, transthyretin in ATTR, and p53 aggregates in p53-mutated cancers (Hoshono-Fukao et al., Am J

Pathol, 2012, PMID: 22429964; Kameyama et al., Am J Pathol, 2019, PMID: 30414409; Iwahashi, PNAS, 2020, PMID: 33318190). Please also see our reply to Comment 5 below. **We performed additional immunohistochemical analysis with the β 0001 anti-A β antibody and the ProteoStat dye that recognizes protein aggregates. The methods have been updated on page 20, lines 389–399. The results are now included in revised Fig. 1 and described on page 5, line 101–page 6, line 108.** During the preparation of our revised manuscript, we noticed that we missed include that we used the TrueVIEW Autofluorescent Quenching Kit for quenching the endogenous autofluorescence derived from red blood cells. **We now included the use of this kit in the methods on page 20, lines 392–395.** We would like to thank the reviewers for providing us with the opportunity to correct our mistake. The original Fig.1 shows that A β was co-deposited with heparan sulfate S-domains. We think that interactions between A β and biomolecules including heparan sulfate are important for A β aggregation and deposition. Accordingly, please kindly be informed that **we keep the original Fig.1 as Supplemental Fig. S1.**

Page 5, line 101–page 6, line 108

We used the ProteoStat Protein Aggregation Assay kit for analysis of aggregated A β . A β peptides are small peptides consisting of 40 to 42 amino acid residues that are released extracellularly after production. Non-deposited A β monomers are not detected by our immunohistochemical analysis, because these soluble A β peptides are spread out in the tissue fluid. Thus, we calculated only merged signals of A β and the ProteoStat dye in order to show aggregated and deposited A β peptides in the placenta. Here, we found that significant deposition of aggregated A β in the villi of five PE cases but not in normal placentas (Figure 1).

Page 20, lines 389–399 (including page 20, lines 392–395 for the TrueVIEW kit)

For detection of A β aggregates, sections were washed with PBS, incubated in ProteoStat solution (1:2,000 in the ProteoStat assay buffer) for 3 min, and then destained in 1% acetic acid for 20 min at room temperature (Matafora et al., 2020). Endogenous autofluorescence derived from red blood cells were quenched by using the TrueVIEW Autofluorescence Quenching Kit (Vector Laboratories) and the sections were mounted with Vectashield mounting medium with 4',6-diamidino-2-phenylindole (DAPI) (Vector Laboratories). Specimens were then examined with an LSM700 confocal microscope (Carl Zeiss, Jena, Germany). For quantification, 5 regions of interest (ROIs; 16 μ m \times 16 μ m) were set on villi in each placenta tissue, and mean intensities were determined by using ZEN 3 blue edition (Carl Zeiss). β 001- and ProteoStat-positive areas were determined by using the Coloc. Tools of ZEN 3.

3. What appeared to be more surprising is the statement on lines 162 and 163 that cultured CTBs produced A β 40/42. Again, it is not clear whether the authors are talking about aggregated A β or just induction of A β . Why should normal CTBs produce A β ? It is not clear whether this is a transient expression or a long-term phenomenon. The issue is distinction between normal and adverse pregnancy conditions, and the latter associated with protein aggregation as suggested in the literature.

Our response: BeWo cells and cultured CTBs produce A β peptides in a normoxic condition. In the brain, neurons constitutively produce A β peptides, which have physiological roles such as controlling neuronal hyperexcitability, enhancing of synaptic plasticity, and improving memory (reviewed in Kent et al., Acta Neuropathol, 2020, PMID: 32728795). The amount of A β in the brain is regulated by the balance between A β production and A β clearance, and the imbalance of the production and the clearance may result in an increase in A β local concentration and A β aggregation. Our results showing that hypoxia increased A β production in BeWo cells suggest that chronic hypoxia, which is a risk of preeclampsia, may lead to a sustained increase in A β production and an elevated local concentration of A β at or near the site of A β production. **As we mentioned on page 5, line 101–page 6, line 108,** we detected A β deposition in PE placentas and non-aggregated A β was not detected because soluble A β peptides in normal placentas are thought to spread out in the tissue fluid. **We discussed these points in the discussion on page 7, lines 129–131.**

In the present study, we showed that aggregated A β (i.e., A β fibrils) was detrimental to the CTB differentiation. On the other hand, we found that A β monomers promoted EVT invasion. We believe that promotion of EVT invasion by A β monomers represent a physiological function of A β in the placenta. **We included the EVT data as newly prepared Supplemental Figures S5 and S9 and discussed on page 8, line 159–page 9, line 164, and page 13, lines 250–258. We also performed experiments with BeWo cells and non-aggregated A β in order to confirm that aggregated A β fibrils are detrimental to CTB syncytialization. These results are included as newly prepared Supplemental Figure S6 and described in page 9, lines 178–180.**

Page 5, line 101–page 6, line 108

We used the ProteoStat Protein Aggregation Assay kit for analysis of aggregated A β . A β peptides are small peptides consisting of 40 to 42 amino acid residues that are released extracellularly after production. Non-deposited A β monomers are not detected by our immunohistochemical analysis, because these

soluble A β peptides are spread out in the tissue fluid. Thus, we calculated only merged signals of A β and the ProteoStat dye in order to show aggregated and deposited A β peptides in the placenta. Here, we found that significant deposition of aggregated A β in the villi of five PE cases but not in normal placentas (Figure 1)

Page 7, lines 129–131

Our immunohistochemical analysis revealed that early- or late-onset PE placentas were in a hypoxic condition, which may lead to a sustained increase in the production and local concentration of A β peptides and fibrillization of A β .

Page 8, line 159–page 9, line 164

Although our ELISA failed to detect A β 40 in the conditioned medium (CM) of the EVT model HTR8/SVneo cells (Graham et al., 1993), we found that HTR8/SVneo cells produced A β peptides and that the A β production was enhanced by hypoxia in the immunoblot with the β 001 antibody (Supplemental Figure S5). These results suggested that HTR8/SVneo cells produced much less amount of A β than BeWo cells.

Page 9, lines 178–180

We observed that non-aggregated A β did not affect the secretion of β -hCG (Supplemental Figure S6), which corroborated the detrimental effects of aggregated A β on CTB syncytialization.

Page 13, lines 250–258

Here, we also found that HTR8/SVneo cells, a widely accepted model of EVT (Graham et al., 1993), produced much less amount of A β than BeWo cells and nM range of A β promoted EVT invasiveness (Supplemental Figure S9). We identified 1444 differentially expressed genes in A β -treated HTR8/SVneo cells and our transcriptome analysis suggested that quite low concentration of A β enhanced EVT invasiveness by activating R-HSA-1442490 (collagen degradation) and R-HSA-1592389 (activation of matrix metalloproteases). These results suggest that while A β fibrils are detrimental to the placenta, nM range of A β monomers may have a physiological function in early pregnancy.

4. The authors have adequately pointed to importance of hypoxia in the onset of preeclampsia-like features. As a matter of fact, Lai et al demonstrated in a mouse pre-clinical model that hypoxia could induce severe features of preeclampsia (*Hypertension*. 2011;57:505-514). The use of hypoxia as driver of A β induction is appreciated.

Our response: We agree with the reviewer that studies using preclinical animal models are an important topic for the future. **We discussed this point in the discussion on page 17, lines 332–337.**

Page 17, lines 332–337

The limitation of this study is that we used a cellular model for demonstrating hypoxia-enhanced A β production. Lai et al. reported a preclinical PE model by coupling low oxygen tension and an interleukin-10-deficiency (Lai, Kalkunte, & Sharma, 2011). An important future research topic will be the use of preclinical models to further investigate the role of hypoxia in placental A β production and the role of A β aggregates in the etiology and pathology of PE.

5. In Fig. 1, although the authors have used DIC approach, it would have been helpful if they presented individual A β and RB4CD12 green and red channels, and a merged profile. For example, PE #4 sample does not appear to have much RB4CD12. Again, there is a question of aggregated or native protein structures. It is difficult to have a satisfactory statistical analysis. Did the authors look for A β in the anchoring villi region of the placenta?

Our response: We showed the green and red channel images individually in Supplemental Figure S1. We have noticed that we detected A β deposition without RB4CD12 signals. A β is small peptides of 40 to 42 amino acid residues and is extracellularly released after the production. Non-deposited A β monomers are not detected by immunohistochemical analysis, because these soluble A β peptides are spread out in the tissue fluid. **We now included immunohistochemical analysis with the anti-A β antibody and the ProteoStat dye as revised Fig. 1.** In our statistical analysis, we calculated only merged signals of A β and ProteoStat or RB4CD12, which suggests that our data show the aggregated and deposited A β . **We described this point in the results on page 5, line 101–page 6, line 113, and the methods on page 20, lines 389–399.** Please also see our response to Comment 2 above. We did not observe A β deposition at the anchoring villi.

Page 5, line 101–page 6, line 113

We used the ProteoStat Protein Aggregation Assay kit for analysis of aggregated A β . A β peptides are small peptides consisting of 40 to 42 amino acid residues that are released extracellularly after production. Non-deposited A β monomers are not detected by our immunohistochemical analysis, because these soluble A β peptides are spread out in the tissue fluid. Thus, we calculated only merged signals of A β and the ProteoStat dye in order to show aggregated and deposited A β peptides in the placenta. Here, we found that significant deposition of aggregated A β in the villi of five PE cases but not in normal placentas (Figure 1). We also used the RB4CD12 anti-heparan sulfate S-domain antibody as an amyloid/protein aggregate marker, because heparan sulfate S-domains have been shown to co-deposit with amyloid *in vivo* (Bruinsma et al., 2010; Hosono-Fukao et al., 2012; Iwahashi et al., 2020; Kameyama et al., 2019; Kazuchika Nishitsuji, 2018; K. Nishitsuji & Uchimura, 2017). Again, we found co-deposition of A β with the RB4CD12 epitope in PE placentas (Supplemental Figure S1).

Page 20, lines 389–399

For detection of A β aggregates, sections were washed with PBS, incubated in ProteoStat solution (1:2,000 in the ProteoStat assay buffer) for 3 min, and then destained in 1% acetic acid for 20 min at room temperature (Matafora et al., 2020). Endogenous autofluorescence derived from red blood cells were quenched by using the TrueVIEW Autofluorescence Quenching Kit (Vector Laboratories) and the sections were mounted with Vectashield mounting medium with 4',6-diamidino-2-phenylindole (DAPI) (Vector Laboratories). Specimens were then examined with an LSM700 confocal microscope (Carl Zeiss, Jena, Germany). For quantification, 5 regions of interest (ROIs; 16 $\mu\text{m} \times 16 \mu\text{m}$) were set on villi in each placenta tissue, and mean intensities were determined by using ZEN 3 blue edition (Carl Zeiss). β 001- and ProteoStat-positive areas were determined by using the Coloc. Tools of ZEN 3.

6. *Fig. 2 does not show significant staining for HIF1- α in PE placental tissue.*

Our response: In a normoxic condition, HIF1- α is constitutively expressed but degraded via the proline-hydroxylation and the subsequent ubiquitination and degradation in the proteasome. Because the proline-hydroxylation is oxygen-dependent, hypoxia induce HIF1- α accumulation. Thus, our data suggest a hypoxic environment in the preeclamptic placentas. **We described this point in the results section on page 6, line 117–page 7, line 123.**

Page 6, line 117–page 7, line 123

Because PE placentas exist under hypoxic conditions (Tong et al., 2022), we analyzed the expression of HIF1- α in human normal and PE placentas. In a normoxic condition, HIF1- α is constitutively expressed but degraded via the proline-hydroxylation and the subsequent ubiquitination and degradation in the proteasome. Because the proline-hydroxylation is oxygen-dependent, hypoxia induce HIF1- α accumulation (Masson & Ratcliffe, 2003). Immunohistochemistry revealed the induction of HIF-1 α in PE placentas, suggesting a hypoxic environment consistent with a previous study (Caniggia & Winter, 2002).

7. *Fig. 3B, why should there be A β 40/42 under normoxic conditions? This is the most pertinent concern and the authors are validating significant expression of A β 40/42 under normal conditions. In normal pregnancy placenta, this protein has not been detected.*

Our response: A β peptides are constitutively produced in BeWo cells, and the production was enhanced by hypoxia. A β is small peptides of 40 to 42 amino acid residues. We did not observe A β signals in the immunohistochemical analysis of the normal pregnancy placentas, because A β peptides that do not aggregate and deposit in the placenta were distributed in the tissue fluid and lost before and during the processing of the placentas for the paraffin-embedding and immunostaining. Our immunohistochemical analysis detects only A β deposition. Thus, the absence of A β signals in the immunohistochemical analysis of normal placentas does not mean that normal placenta does not produce any A β peptides. **We described this point on page 6, lines 103–108, and on page 7, lines 133–135.**

Page 6, lines 103–108

Non-deposited A β monomers are not detected by our immunohistochemical analysis, because these soluble A β peptides are spread out in the tissue fluid. Thus, we calculated only merged signals of A β and the ProteoStat dye in order to show aggregated and deposited A β peptides in the placenta. Here, we found that significant deposition of aggregated A β in the villi of five PE cases but not in normal placentas (Figure 1).

Page 7, lines 133–135

Although we detected BACE1 in normal placentas, A β did not deposit in these placentas. As mentioned above, non-deposited soluble A β peptides in normal placentas are thought to spread out in the tissue fluid and were not detected by our immunohistochemical analysis.

8. Figs. 4 and 5 present the crux of the conclusions that the authors are trying to draw from their study. A β peptide solution was incubated for 5 days at 37°C to prepare so called A β fibril-like structures. What is the purity of fibril structures? Does this preparation show toxic effects on cell viability? Human trophoblasts expressing E-cadherin fail to participate in endovascular cross-talk with endothelial cells, a process required for spiral arteries. It appears that either BeWo cells or primary trophoblasts used in this study represent trophoblasts from third trimester. It is not clear why should A β fibril like structures should inhibit ZO-1 and E-cadherin or β -hCG (Fig. 5) for that matter. In Fig. 5C, there does not seem to be a major effect of A β fibrils. Did the authors try synthetic A β as a control. These experiments could have been meaningful but for proper controls.

Our response: Synthetic A β was purchased from Peptide Institute (Osaka, Japan). The purity is >95%. **We included the data sheet as a review process file.** In case that the reviewer wants to know the fibril content of the preparation, **we confirmed that our A β fibril preparation contained any non-aggregated A β by using Native PAGE followed by Western blotting and dot blotting with the OC anti-amyloid fibril antibody. The results are shown as newly prepared Supplemental Figure S10c and mentioned this point on page 22, lines 427–429.** We did not observe any cytotoxicity of the preparation as shown in Supplemental Fig. S8.

We previously showed that membrane localization of cell-cell interaction proteins such as ZO-1 and E-cadherin in cytotrophoblasts is required for syncytialization (Iwahashi et al., Endocrinology, 2019, PMID: 30551188; Matsukawa et al., Biomolecules, 2022, PMID: 36008943). Because A β aggregates disrupt membrane localization of tight junction proteins partly by inducing excess autophagy (Marco et al.,

Neurosci Lett, 2006, PMID: 16644119; Chan et al., Exp Cell Res, 2018, PMID: 29856989), we hypothesized that A β fibrils may also disrupt membrane localization of ZO-1 and E-cadherin in BeWo cells. We are focusing on the effect of A β fibrils on cytotrophoblasts at the late stage of pregnancy when the remodeling of spiral arteries is completed. Thus, we do not look at the endovascular crosstalk between trophoblasts and endothelial cells. **We described this point on page 10, lines 182–187, and on page 13, lines 258–260.** We agree with the reviewer in that understanding roles of the effects of A β and A β fibrils on early pregnancy is a very important research topic. **We cited an article showing the effects of A β aggregates on EVT β s (Gao et al., J Mol Histol, 2024, PMID: 38777993) and included our new data showing the A β monomer functions on EVT invasion as newly prepared Supplemental Figures S5 and S9. We also noted these points on page 13, lines 243–258.** Please also see our reply to Comment 3 above. **As for Fig. 5C, our improved images now clearly show the loss of membrane localization of ZO-1 and E-cadherin. We also included the results showing that non-aggregated A β did not affect syncytialization of BeWo cells. These results are included as newly prepared Supplemental Figure S6 and noted on page 9, lines 178–180.**

Page 9, lines 178–180

We observed that non-aggregated A β did not affect the secretion of β -hCG (Supplemental Figure S6), which corroborated the detrimental effects of aggregated A β on CTB syncytialization.

Page 10, lines 182–187

A β aggregates disrupt membrane localization of tight junction proteins, at least in part, by inducing excess autophagy (Chan et al., 2018; Marco & Skaper, 2006). We hypothesized that A β fibrils might also disrupt the membrane localization of ZO-1 and E-cadherin in cytotrophoblasts. Therefore, we next investigated the effect of A β 42 fibrils on the subcellular localization of ZO-1 and E-cadherin in BeWo cells and how this affects CTB syncytialization.

Page 13, lines 243–258

A β aggregates reportedly enhanced autophagy in HTR8/SVneo cells and an excessive autophagy may exacerbate PE (L. Gao et al., 2015; Q. Gao et al., 2024). On the other hand, immediately after the implantation, an embryo has to survive a severe low oxygen tension condition due to the lack of vasculature and oxygen supply (Pollheimer, Vondra, Baltayeva, Beristain, & Knofler, 2018; Rodesch, Simon, Donner, & Jauniaux, 1992). Thus, EVT β s aggressively invade the uterine decidua and myometrium to remodel the maternal spiral arteries and develop a vasculature for the maternal-fetal interface (Pijnenborg, Dixon, Robertson, & Brosens, 1980; Tayade, Black, Fang, & Croy, 2006). Here,

we also found that HTR8/SVneo cells, a widely accepted model of EVT_s (Graham et al., 1993), produced much less amount of A β than BeWo cells and nM range of A β promoted EVT invasiveness (Supplemental Figure S9). We identified 1444 differentially expressed genes in A β -treated HTR8/SVneo cells and our transcriptome analysis suggested that quite low concentration of A β enhanced EVT invasiveness by activating R-HSA-1442490 (collagen degradation) and R-HSA-1592389 (activation of matrix metalloproteases). These results suggest that while A β fibrils are detrimental to the placenta, nM range of A β monomers may have a physiological function in early pregnancy.

Page 13, lines 258–260

Elucidation of the effects of A β or A β fibrils in the endovascular crosstalk between trophoblasts and vascular endothelial cells is a future research topic.

Page 22, lines 427–429

We also determined the fibril content to be approximately 94% through analysis with native-PAGE, followed by Western blotting with β 001 and dot blotting with the OC anti-amyloid antibody (Kayed et al., 2007) (Supplemental Figure S10c).

[All figures have been removed by editorial staff per authors' request].

December 24, 2025

RE: Life Science Alliance Manuscript #LSA-2025-03453R

Dr. Kazuchika Nishitsuji
Wakayama Medical University
811-1 Kimiidera
Wakayama 6418509
Japan

Dear Dr. Nishitsuji,

Thank you for submitting your revised manuscript, "Amyloid- β fibrils accumulated in preeclamptic placentas suppress cytotrophoblast syncytialization" to LSA. We apologise for the delay in communicating our decision only now (due to editor availability issues and previous delays in securing reviewer comments).

Your revised manuscript was evaluated by two of the original reviewers whose comments are appended below. As you will read, the reviewers are consistent in their views that the revised manuscript has satisfactorily addresses their previous concerns.

In line with the reviewers' evaluation, we would be happy to publish your paper in Life Science Alliance pending final revisions necessary to meet our formatting guidelines.

- In the methods section, please, provide a citation (if applicable) for the diagnostic criteria for PE as determined by the International Society for the Study of Hypertension in Pregnancy.
- include imaging details (type of objective used, magnification and NA)
- remove the primer information (RT-PCR) from the legend of figure S7 and include it in the respective methods section for RT-PCR experiments.
- Kindly include a scale bar and respective scale bar information in the figure legend for Figure 4B (high magnification images) and Figure S9A.
- Please incorporate the supplementary methods description provided as a separate file into "Methods' section of the main text.
- Please remove a separate supporting information file.
- Please add the X and Bluesky handles of your host institute/organisation, as well as your own and/or one of the authors, in our system.
- Please incorporate supplementary references into the main references list.
- Please add callouts for Figure S9A-C to your main manuscript text.
- please be sure that the authorship listing and order is correct

LSA now encourages authors to provide a 30-60 second video where the study is briefly explained. We will use these videos on social media to promote the published paper and the presenting author (for examples, see <https://docs.google.com/document/d/1-UWCfbE4pGcDdcgzcmiuJl2XMBJnxKYeqRvLLrLS08s/edit?usp=sharing>). Corresponding or first-authors are welcome to submit the video. Please submit only one video per manuscript. The video can be emailed to contact@life-science-alliance.org

A. FINAL FILES:

- An editable version of the final text (.DOC or .DOCX) is needed for copyediting (no PDFs).
- High-resolution figure, supplementary figure and video files uploaded as individual files: See our detailed guidelines for

preparing your production-ready images, <https://www.life-science-alliance.org/authors>

B. MANUSCRIPT ORGANIZATION AND FORMATTING:

Thank you for your attention to these final processing requirements. Please revise and format the manuscript and upload materials as soon as you are able.

Sincerely,

Sarita Hebbar, PhD
Scientific Editor
Life Science Alliance
<http://www.lsajournal.org>

Reviewer #1 (Comments to the Authors (Required)):

Compared to the previous version, the authors answered to some of my questions, and justify the absence of some experiments that I suggested such as di-8 fluorescent labeling to quantify cell fusion. I think that the extent of the response to concerns of the other reviewers is also significant, such as the RNAseq on EVT models.

Reviewer #3 (Comments to the Authors (Required)):

The authors have made good effort to address the concerns raised so that the conclusions are strengthened.

January 2, 2026

RE: Life Science Alliance Manuscript #LSA-2025-03453RR

Dr. Kazuchika Nishitsuji
Wakayama Medical University
811-1 Kimiidera
Wakayama 6418509
Japan

Dear Dr. Nishitsuji,

Thank you for submitting your Research Article entitled "Amyloid- β fibrils accumulated in preeclamptic placentas suppress cytotrophoblast syncytialization". It is a pleasure to let you know that your manuscript is now accepted for publication in Life Science Alliance. Congratulations on this interesting work.

DISTRIBUTION OF MATERIALS:

Again, congratulations on a very nice paper. I hope you found the review process to be constructive and are pleased with how the manuscript was handled editorially. We look forward to future exciting submissions from your lab.

Best wishes for a happy new year,

Sarita Hebbar, PhD
Scientific Editor
Life Science Alliance
<http://www.lsjournal.org>